


# Reconstruction and simulation of an extreme flood event in the Lago Maggiore catchment in 1868

Peter Stucki[1,2], Moritz Bandhauer[1,2,a], Ulla Heikkilä[3,b], Ole Rössler[1,2], Massimiliano Zappa[4], Lucas Pfister[2], Melanie Salvisberg[1,5], Paul Froidevaux[2,3], Olivia Martius[1,2], Luca Panziera[1,2,6], Stefan Brönnimann[1,2]

[1]Oeschger Centre for Climate Change Research, University of Bern, Bern, 3012, Switzerland
[2]Institute of Geography, University of Bern, Bern, 3012, Switzerland
[3]Meteotest, Bern, 3012, Switzerland
[4]Eidg. Forschungsanstalt WSL, Birmensdorf, 8903, Switzerland
[5]Institute of History, University of Bern, Bern, 3012, Switzerland
[6]MeteoSvizzera, Locarno Monti, 6605, Switzerland
[a]now at Schweizerische Energie-Stiftung SES, Zurich, 8005, Switzerland
[b]now at LogObject, Zurich, 8048, Switzerland

*Correspondence to*: Peter Stucki (peter.stucki@giub.unibe.ch)

**Abstract.** Heavy precipitation on the south side of the central Alps produced a catastrophic flood in October 1868. We assess the damage and societal impacts, as well as the atmospheric and hydrological drivers using documentary evidence, observations, and novel numerical weather and runoff simulations.

The greatest damage was concentrated close to the Alpine divide and Lago Maggiore. An atmospheric reanalysis emphasizes the repeated occurrence of streamers of high potential vorticity as precursors of heavy precipitation. Dynamical downscaling indicates high freezing levels (4000 m a.s.l.), extreme precipitation rates (max. 270 mm/24 h), and weather dynamics that agree well with observed precipitation and damage, and with existing concepts of forced low-level convergence, mid-level uplift and iterative northeastward propagation of convective cells. Simulated and observed peak levels of Lago Maggiore differ by 2 m, possibly because the exact cross-section of the lake outflow is unknown. The extreme response of Lago Maggiore cannot be attributed to low forest cover. Nevertheless, such a paradigm was adopted by policy makers following the 1868 flood, and used to implement nationwide afforestation policies and hydraulic structures.

These findings illustrate the potential of high-resolution, hydro-meteorological models – strongly supported by historical methods – to shed new light on weather events and their socio-economic implications in the 19th century.

*Keywords*: reanalysis, dynamical downscaling, weather model chain, historical flood event, Alps, heavy precipitation, historical climatology

## 1    Introduction

Floods are natural hazards with potentially disastrous impacts on built structures and landscapes: In many parts of Europe and the Alps, they represent the most damaging and expensive (hydro-) meteorological hazard (Hilker et al., 2009). A number of heavy precipitation events producing severe floods have occurred on the southern side of the Central Alps (AS hereafter) in recent decades. The most extreme of these recent events occurred in October 2000 and in September 1993 (Bundesamt für Wasser und Geologie, 2002; Grebner, 1993, 2000; Stucki et al.,



2012). These events have been analyzed thoroughly in several studies and administrative reports, e.g. regarding impacts, damages and hydro-meteorological processes (Boudevillain, 2009; Bundesamt für Wasser und Geologie, 2002; Buzzi et al., 1998; Buzzi and Foschini, 2000; Cassandro Cremonini et al., 2001; Fehlmann et al., 2000; Grebner, 1993, 2001; Massacand et al., 1998; Röthlisberger, 1994; Rotunno and Houze, 2007; Schlemmer et al., 2010; Stucki et al., 2012, 2013; Tonzani and Troisi, 1993).

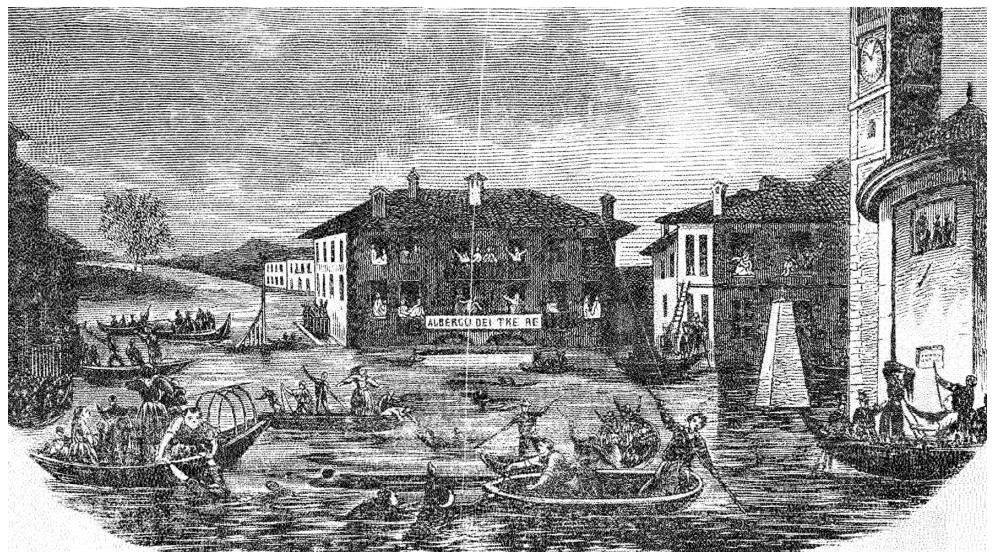

**Figure 1. Engraving showing the historical center of Sesto Calende, located at the southern end of Lago Maggiore (see label in Figure 2), during the 1868 flood event. Taken from Bogni (1869).**

These two extreme floods were perceived as 'centennial', with unpredictable impacts (e.g. Bundesamt für Wasser und Geologie, 2002). However, they also led to a scientific, political, and societal re-discovery of the most disastrous flood event on the AS in centuries: the extreme flood from 27 September 1868 to 4 October 1868 on Lago Maggiore and adjacent regions (LMR hereafter; Figure 1; see also labels in Fig. 2). For instance, Maugeri et al. (1998) provided a reconstruction of air pressure, temperature, wind, and cloudiness over Italy for 3 October 1868, based on sparse meteorological observations. Petrascheck (1989) collected and mapped the flood damages and compared them to a severe flood in 1987. Pfister (1999, 2002), Pfister and Summermatter (2004) and Summermatter (2005, 2007, 2012) analyzed the societal impacts and political ramifications based on documentary sources such as the hydro-meteorological and impact reports by Coaz (1869) and Arpagaus (1870). They found that the event had far-reaching consequences regarding flood protection measures in Switzerland. In reaction to the catastrophic flood, the federal government created the legal requirements to coordinate and subsidize protection measures nationwide. In the decades following the 1868 flood, numerous watercourses were corrected and large areas of alpine meadows reforested. In general, the scientific reports found that the 1868 flood occurred during a period of almost persistent precipitation between 13 / 14 September 1868 and 9 October 1868; and only from 4 October 1868, relevant precipitation amounts fell as snow at higher elevations (Coaz, 1869).



Such traditional reconstructions are typically based on documentary evidence and observations. Although remarkable achievements can be made with these methods, it is desirable to complement the results with highly resolved spatial and temporal information, which is needed for applications like risk assessments and damage
4    modelling.

In a first step, we aim to extend existing knowledge using recently digitized hydro-meteorological observations from national projects (Füllemann et al., 2011) and from our own research. We also aim to provide a better understanding of the ensuing policy regarding natural hazards in Switzerland, using recent historical studies.
8    Specifically, we describe the damage extent, intensities and societal impacts, and reconstruct the flood and the associated precipitation using available observations.

In a second step, we make use of a global atmospheric reanalysis dataset that has recently been extended to the year 1851. This dataset facilitates analyses of the large-scale (synoptic) weather conditions during the 1868 flood
12   and allows for dynamical downscaling, i.e. the nesting of a limited-area weather model, which in turn provides highly resolved information for local-scale weather analyses (cf. Michaelis and Lackmann, 2013 for a blizzard simulation in 1888; Welker et al., 2015 for historic high-impact windstorms). We analyze hydro-meteorological atmospheric conditions (e.g. moisture flux, precipitation rates, the height of the freezing level) on synoptic and
16   meso-scales, and we compare the model output to the observations and historical sources.

In a third step, we make use of a novel analog sampling method for hydro-meteorological simulations (Rössler and Brönnimann, 2018). Specifically, we explore the use of the simulated precipitation and other variables (including the role of forested areas) to assess the hydrological response of the Lago Maggiore water level.
20   As in many mountainous regions, typical hydro-meteorological ingredients associated with heavy and sustained precipitation over the central Alps include conditionally unstable layers in the troposphere, sustained and intense advection of moist low-level air masses, and an orographically forced ascent (Doswell et al., 1996; Lin et al., 2001).
24   On a synoptic scale, extreme precipitation events on the AS are linked to upper-level troughs, i.e. regions of deep layers of relatively cool air masses located over (south-)western Europe (e.g., Boudevillain, 2009; Buzzi and Foschini, 2000; Stucki et al., 2012). In a so-called PV-perspective, they correspond to the presence of meridionally elongated intrusions of stratospheric air, which denote streamers of high potential vorticity (PV) extending
28   southwards from polar regions in the Northern Hemisphere (Appenzeller and Davies, 1992; Martius et al., 2006; Massacand et al., 1998). A number of studies have identified such PV-streamers over western Europe as precursors of heavy precipitation events on the AS (Fehlmann et al., 2000; Hoinka et al., 2006; Massacand et al., 1998). Typically, these streamers are accompanied by reduced static stability and by a southerly and ascending flow of
32   moist air on their eastern side. Hence, a PV-streamer located west of the Alps will likely produce a large moisture flux (hereafter represented by the vertically integrated water vapor transport; IVT) towards the Alps, which is a necessary ingredient for heavy precipitation and floods over the AS (Froidevaux and Martius, 2016; Hoinka et al., 2006; Martius et al., 2006; Schlemmer et al., 2010). Barton et al. (2016) found that the rapid succession of heavy
36   precipitation events in the LMR can lead to floods. They also found that the temporal clustering of regional extreme precipitation on sub-seasonal time scales in the LMR is statistically significant.

On smaller scales, the complex orography of the AS further modulates the moisture flow, potentially generating heavy rainfall (Gheusi and Davies, 2004; Panziera and Germann, 2010; Rotunno and Houze, 2007). The AS is
40   particularly prone to severe floods due to its characteristic topography, with large elevation differences, steep





slopes and the concave form of the Alpine barrier, which forces southerly moist air flow to converge, rise, and condensate (e.g. Frei and Schmidli, 2006; Schneidereit and Schär, 2000). For flash floods in the LMR, Panziera et al. (2015) found repeated northeastward propagation of convective cells that produce a spatially elongated area with very large precipitation amounts (for a hydrological application, see Liechti et al., 2013). At near-surface levels, moist and conditionally unstable air is advected from (south-)easterly directions, rises to condensation over the slopes of the LMR, and is then captured, carried on and lifted in southerly to southwesterly flow at higher levels. In addition, Rotunno and Houze (2007; and references therein) noted cases where forced uplift in a strong cross-barrier flow controls much of the condensation, but where intermittent convective cells also occur despite rather low values of convective available potential energy (CAPE). Such a typical meso-scale configuration may persist for a period of up to a couple of days.

Once the precipitation falls, the hydrological response in the LMR is quick, due to the steep terrain, shallow soils, and a land cover that is approximately 40 % non-forested (Andres et al., 2016). Very high runoff can form within hours, resulting in high runoff variability for AS river systems. Those rivers (i.e. Ticino, Toce, Maggia, Verzasca, and Moesa) all discharge into the Lago Maggiore, which – at least to some degree – has a dampening effect on the flood wave propagation to downstream areas. At the same time, this strong concentration of river systems leads to a high disposition of the LMR to lake level rise and flooding. One advantage of the steep topography relates to the zero-degree line that is likely present during colder storm events, resulting in snow deposit rather than direct runoff. However, these snow deposits can also amplify the runoff volume during subsequent rainfall events. We will look at the presence and representation of all these processes in historical data, a reanalysis product, and a high-resolution simulation covering the 1868 event. The study is organized as follows. Observational data and models are presented in Sect. 2. Results and discussion regarding damage and societal impacts, the chronology of the heavy precipitation, the simulated atmospheric conditions and the hydrological response of the LMR catchment are all provided in Sect. 3. A summary and conclusions are given in Sect. 4.

## 2    Observational data and models

### 2.1    Meteorological and hydrological observations

Digitized series of daily rates of precipitation (5:40 AM to 5:40 AM of the next day in mm) are provided by the Swiss Federal Office of Meteorology and Climatology MeteoSwiss (Füllemann et al., 2011). Additional observations are digitized from the Annals of the Schweizerische Meteorologische Zentralanstalt (Wolf, 1868), the precursor organization of MeteoSwiss, and from (Coaz, 1869), although some information is redundant. Information about two Italian locations (Pallanza and Como) come from Luino et al. (2005).

Information about historical water levels is taken from Di Bella (2005) and from Stucki and Luterbacher (2010). The first series has sporadic estimations of annual maxima of the Lago Maggiore water level reaching back to 1177; the second goes back to 1500 and includes categorical information on severe or moderate floods. Systematic measurements of annual maxima of the Lago Maggiore water level are available for Sesto Calende since 1839 (taken from Di Bella, 2005), and for Locarno since 1868 (provided by the Swiss Federal Office for the Environment FOEN).





### 2.2    Global and regional models, dynamical downscaling

The Twentieth Century Reanalysis dataset (Compo et al., 2011) version 2c (20CR in the following) is used for synoptic-scale analyses of the 1868 flood. 20CR is a global, three-dimensional atmospheric dataset that is mainly based on surface pressure observations from the International Surface Pressure Databank (ISPD; Cram et al., 2015) version 3.2.9. Data assimilation is performed with an ensemble Kalman filter. In this study, we consider an ensemble mean of 56 members. 20CR has a spatial grid size of 2° latitude x 2° longitude, 24 pressure levels in the vertical, and a temporal resolution of 6 hours. 20CR reaches back to 1851 and is the only global reanalysis to cover the 1868 flood event.

Regional simulations of flood-inducing weather are done by dynamical downscaling. Dynamical downscaling is a method used to obtain regional weather and climate information by nesting regional climate or weather models into global-scale models. That is, the global model drives the regional model from the regional model domain boundaries and for the temporal initialization (Dickinson et al., 1989; Giorgi et al., 1990; Giorgi and Bates, 1989). Dynamical downscaling is employed over the core period of the 1868 flood event between 26 September 00 UTC and 5 October 00 UTC 1868. The regional model used is the Weather Research and Forecasting model WRF-ARW (Advanced Research WRF; Skamarock et al., 2008). The downscaling from 20CR is performed in four nested model domains with grid sizes of 54 km, 18 km, 6 km and 2 km. All domains contain 40 vertical levels. The time step is 11 seconds in the innermost domain and output frequency is one hour. With regard to physics parametrizations, we use Ferrier (eta) for microphysics, Kain Fritsch for convection, Noah for the land surface, revised MM5 for the surface layer, YSU for the planetary boundary layer and Dudhia and RRTM for short- and long-wave radiation (see the WRF ARW user guide at http://www2.mmm.ucar.edu/wrf/users/docs/user_guide_V3/contents.html). The convection parametrization is turned off for the innermost domain because deep convection is resolved explicitly by the model at this scale. In order to maintain the large-scale information in WRF close to 20CR during the entire simulation, we apply spectral nudging to horizontal wind, temperature and geopotential height in the outermost domain. We nudge only wavelengths >1000 km with a relaxation time scale of one hour. Nudging is switched off in the planetary boundary layer and starts from the 10[th] model level upwards.

### 2.3    Analog resampling method and hydrological model

An analog resampling method (see Flückiger et al., 2017; Rössler and Brönnimann, 2018, for details) is used to reconstruct daily weather patterns of precipitation and temperature during the 12 months before the extreme flood in early October 1868. With this method, we select analog days in the modern period from 1961 to 2015 that are most similar (in terms of observed pressure fields, temperature and precipitation amounts) to the historical period between 1 October 1867 and 31 October 1868. For physical consistency, analog days must have the same weather type and be within the same season as the days to be reconstructed. Here, the CAP7 weather types from Schwander et al. (2017) are used. The analog selection is based on station data from Lugano and Zurich provided by MeteoSwiss and the instrumental series of Milano and Padova, taken from the IMPROVE project (Camuffo and Jones, 2002). Additionally, time series of precipitation and temperature are extracted from 20CR at the grid point 6° E 46° N. The data are standardized and the Euclidean distance is applied as a similarity measure for analog





days. For the best analog day, we extract temperature and precipitation from the MeteoSwiss RhiresD 2.2 km gridded dataset (Frei, 2014; Frei et al., 2006; Frei and Schär, 1998), as well as from the E-OBS dataset (Haylock et al., 2008). These two products are combined to obtain a full meteorological input for the greater LMR. For

4   precipitation, we find strong deviations between E-OBS and the Swiss precipitation grid for the overlapping area in Switzerland. This topography-induced bias is compensated for by adding an empirically determined 30 % of precipitation for the E-OBS grid cell. Temperature values are not adjusted.

The hydrological simulations in this study are accomplished using the semi-distributed rainfall-runoff model

8   PREVAH (Gurtz et al., 1999; Viviroli et al., 2009), previously calibrated for the LMR hydrological system until the Ticino–Sesto Calende (SC in Fig. 2) gauge at the outflow of the Lago Maggiore (Andres et al., 2016). The original model version by Andres et al. (2016) runs at hourly time steps and considers the numerous regulations and hydro-power plants present today by coupling with a routing model RS (Jordan et al., 2012). As we are

12   interested in the unregulated hydrological system of 1868, we simulate removed regulations by only applying the hydrological model. The entire catchment is subdivided into 37 headwater catchments that are routed using a simplified scheme, aggregating runoffs to daily sums. This approach seems valid because the concentration time for all catchments upstream of the lake is shorter than one day. For the semi-distributed hydrological model, the

16   meteorological fields inform 100 m elevation bands in each sub-catchment. For this project, the data from Arealstatistik 1992 / 1997 (south Ticino, available at https://www.bfs.admin.ch/bfs/de/home/statistiken/raum-umwelt/erhebungen/area/geschichte/area-1992-97.html) and Arealstatistik 2004 / 2009 (North Ticino; available at https://www.bfs.admin.ch/bfs/de/home/statistiken/raum-umwelt/erhebungen/area/geschichte/area-2004-09.html)

20   are used. The domain in Italy is covered by CORINE Land Cover data 2006 (taken from https://www.eea.europa.eu/data-and-maps/data/clc-2006-raster-4). The outflow of Lago Maggiore is displayed by a lake level–outflow relationship that is derived from recent observations and riverbed topography. These observations do not represent the discharge and lake-level conditions in 1868, as a regulatory dam has been

24   operational since 1943. To validate the quality of this adjusted model, we compare simulated against observed lake-levels for the time period 1970–2010 (Figure A1 in the Appendix). Today, Lago Maggiore is heavily regulated. Especially during winter, water is retained to provide irrigation water for downstream lowlands. Consequently, the comparison of long-term means (1980–2010) shows a good representation of lake levels for the

28   summer. For the winter months (November–April), simulations only match when considering a temporal storage of water. As true regulations are unknown, we assume a monthly storage factor between 5 % and 12 % of the inflow.

### 3     Results and discussion

32   #### 3.1     Spatial extent and intensities of damage, societal impact

Information regarding damage locations and intensity, as well as about the weather during the flood event, mainly comes from two contemporary reports (Arpagaus, 1870; Coaz, 1869). They focus on damage in the Swiss Canton of Graubünden in the eastern part of Switzerland, and particularly on flooding along the Alpine Rhine between its

36   sources and Lake Constance (see labels in Fig. 2). However, some damage information for other Cantons is also





available from these sources. The financial information from these reports coincides with the official damage lists found in the Swiss Federal Archives (Schweizerisches Bundesarchiv, 1868a, 1868b, 1868c).

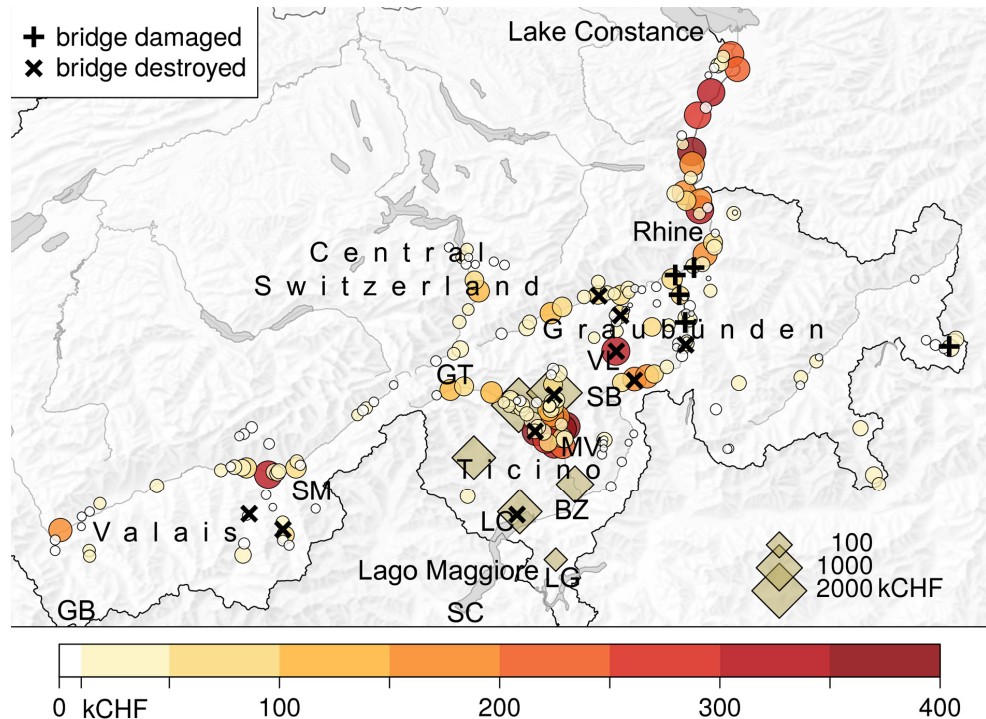

**Figure 2. Damage amounts (in kCHF) as collected from contemporary surveys in 5 Cantons in south-eastern Switzerland. Filled circles (diamonds) refer to damage amounts per municipality (administrative region), where available. Note that both regional and municipality-level values are plotted where possible. Crosses indicate reported damage to bridges. Labels refer to locations mentioned in the text: BZ Bellinzona, GB Grand St. Bernard, GT Gotthard, LC Locarno, LG Lugano, MV Malvaglia, SB San Bernardino, SM Simplon, VL Vals.**

Figure 2 maps the reported damage intensities in terms of Swiss Francs (CHF) on the municipality or regional administrative level where available, and includes information on damaged or destroyed bridges in the area. Overall, the flood of 1868 resulted in 14 million CHF in damage, of which 6.5 million CHF accrued in the Canton of Ticino. In the year 2000, these numbers would be equivalent to 915 million and 425 million CHF, respectively (Summermatter, 2012); that is, they would have increased by a factor of 65. Along with material losses, a high number of fatalities occurred: In total, 50 people were killed by the floods and landslides, 42 of whom were in the Canton of Ticino (Landolt, 1869). One of the most affected municipalities was Malvaglia (label MV in Fig. 2), where 20 people lost their lives and the damages cost an estimated 400'000 CHF. The highest damage amounts were found in the northern part of the Canton of Ticino, with values between 100'000 and 400'000 CHF per municipality, summing up to approx. 4.1 million CHF for the two related administrative regions. The reports indicate that nearly every bridge in Locarno (label LC) was damaged, adding >500'000 CHF in costs in this





municipality alone. In the Canton of Graubünden, nearly all municipalities along the River Rhine were affected by flooding, and several very severe mud flows occurred. For example, the village of Vals (label VL) was virtually covered with mud and damage amounted to around 350'000 CHF. Further downstream, water masses accumulated such that four temporary lakes developed near Lake Constance, leading to damage amounts of approximately 1.2 million CHF in six municipalities. In addition, the breakdown of transport infrastructure had severe ramifications. Many bridges were completely destroyed along the Rhine. Although the Alpine valleys in the easternmost part of Switzerland were spared, severe costs accrued in the Canton of Valais (e.g. with >300'000 CHF in costs along the main valley of the Rhone and to the south) and in several Cantons in Central Switzerland.

Of the overall damages of 14 million CHF, more than half occurred on cultivated land – and therefore had to be borne by private individuals. The water and bedload not only destroyed the harvests, but also made it impossible to cultivate the fields for years. This endangered the livelihoods of the predominantly agricultural population in the poor Alpine regions. As no insurance existed at that time, affected people were reliant on the generosity of their neighbors and fellow citizens. In the days and weeks following the flood event, numerous collections of donations took place, coordinated by the Swiss federal government. They turned out to be a huge success: Pfister (2003) estimates that more than half of all households in Switzerland participated by contributing a large quantity of natural produce and a total of 3.6 million CHF – more than ever before.

Given the extreme magnitude of the damages, the question of the cause of the flood and of protection measures came up immediately. The federal government assigned Carl Culmann, Elias Landolt, and Arnold Escher von der Linth, professors of engineering, forestry, and geology, respectively (all three at the Swiss Federal Institute of Technology in Zürich), to furnish an expert opinion. The same three experts had already delivered reports on potential flood protection measures in the early 1860s. In these reports, they criticized the selective protection methods and, amongst other things, asserted that deforestation in the upper reaches was responsible for the floods in the valley floors (Culmann, 1864; Landolt, 1862). This explanation was not new: First stated by French engineers in the late 18[th] century, the theory that deforestation resulted in more severe flooding had spread widely. In Switzerland, where large areas were deforested as populations expanded during the late 18[th] and early 19[th] century, the connection between floods and deforestation also seemed a plausible explanation for the frequent and severe flood events (e. g. in the years 1834, 1839, and 1852). Lobbying by the Swiss Forestry Society contributed significantly to the dissemination and success of this idea at the federal level (Pfister and Brändli, 1999).

In their reports, dated 1862 and 1864, respectively, the three experts spoke in favor of establishing forest protection measures, a federal legislation and subsidies for river corrections and reforestation projects. Although the Federal Council took note of the statements, they did not take action until the flood of 1868. The extent of the event showed the urgency for protection measures and coordinated action on a national level. In their second assessment, Landolt, Culmann and Escher von der Linth noted that the large amount of precipitation and the saturated soils were the main reason for the flooding. However, they also pointed out the poor conditions of the forests and the insufficient hydraulic structures. Accordingly, they emphasized their recommendations and presented more detailed concepts. The few critical views questioning the generalized correlation between forests and floods (e.g. Blotnitzky, 1869) were not discussed publicly because the Swiss Forestry Society feared losing political momentum (Summermatter, 2012).



This time, the government heeded the proposals of Culmann, Landolt and Escher von der Linth. Only three years after the catastrophic flood, the parliament accepted a federal decree on subsidization for protective structures and reforestation measures. The consequential laws – the Forest Act and the Hydraulic Engineering Act – came into effect in 1876 and 1877, respectively. With these laws, the federal state committed to financially supporting flood protection measures. This had far-reaching impacts: In the following decades, numerous rivers and streams were modified and hundreds of hectares were afforested.

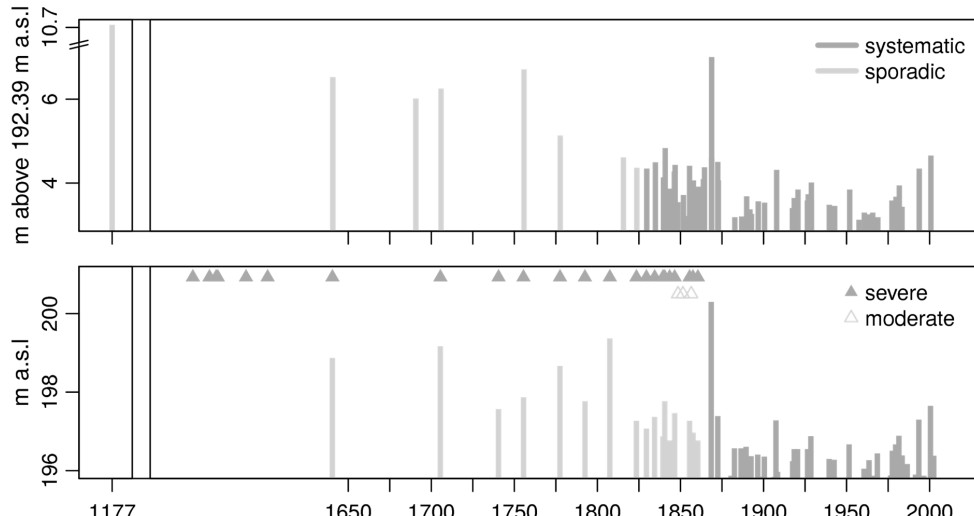

**Figure 3. Reconstructed (light grey bars) and measured (dark grey bars) lake levels at Sesto Calende (top; data from Di Bella, 2005) and Locarno / Lago Maggiore (bottom; data from Stucki and Luterbacher, 2010). In addition, reconstructed flood events classified as 'severe' (dark grey triangles) and 'moderate' (light grey triangles) are shown for Lago Maggiore.**

### 3.2     Reconstruction of water level and precipitation from observations

The water levels of the Ticino river at the outflow from Lago Maggiore in Sesto Calende (label SC in Fig. 2) have been recorded systematically since 1829, and a number of reconstructed water levels are available for the Lago Maggiore area (Figure 3). It can be inferred from the available information that water levels of the Lago Maggiore during the 1868 flood event were arguably the highest since 1177. They are by far the highest in the instrumental period, exceeding the flood levels of October 2000 and September 1993 by >2.5 m (Stucki et al., 2012, 2013; and references therein). In Sesto Calende, flood levels 6.9 m above the zero-measure level were observed.





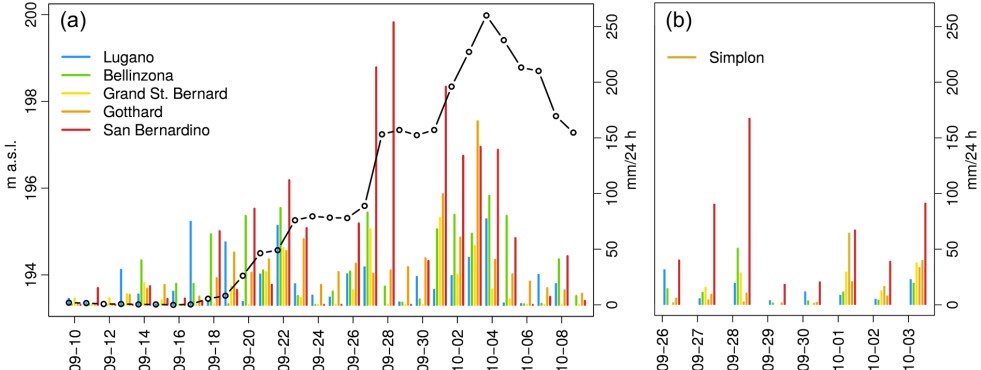

**Figure 4. (a) Mean daily lake level of Lago Maggiore (m a.s.l.; black line) and observed daily precipitation (mm/24 h; from 05:40 local time to 05:40 local time of the next day) at five rain gauge stations between 10 September 1868 and 10 October 1968. (b) Simulated daily precipitation (mm/24 h; from 06 UTC to 06 UTC of the next day) at the locations of (a) plus Simplon Pass between 26 September 1868 and 4 October 1868. Note that the time axis and spacing differ from those in (a).**

Measurements at Locarno show a gradual increase in the water level (daily mean; Figure 4a) from below 194 m a.s.l. before 19 September 1868 to approximately 195 m a.s.l. in the period of 23–26 September 1868, to about 197 m a.s.l. around the end of September 1868. Over the period 3–5 October 1868, water levels peaked at values above 199 m a.s.l. These marked increases were associated with four distinct periods of heavy precipitation on 17–20 September 1868, on 22–23 September 1868, on 26–28 September 1868, and on 1–5 October 1868 (e.g. shown by >50 mm/24 h at >=1 gauge station in Fig. 4). In the following, these periods are called episodes 09-17, 09-22, 09-27, and 10-03, respectively. The measurement stations in the lowland LMR (Lugano, Bellinzona) and on the western part of the Swiss Alpine divide (Grand St. Bernard pass; see labels in Fig. 2) recorded precipitation rates of up to 100 mm/24 h during all periods. The gauge stations on the central and eastern parts of the Swiss Alpine divide reached values of >150 mm/24 h (at Gotthard pass; label GT in Fig. 2) or even up to >250 mm/24 h (San Bernardino pass on 28 September 1868; label SB in Fig. 2). All available observations of daily precipitation amounts are mapped in Fig. 5 for two days within the episodes 09-27 and 10-03, respectively. On 27 September 1868, heavy precipitation was mostly restricted to the AS. One day later, heavy precipitation reached across the Alpine divide in southeastern Switzerland and achieved higher values than elsewhere the day before. Large differences in precipitation amounts over short distances persisted, and no dependence of precipitation intensity on elevation was observed. This indicates convective and potential thunderstorm activity over the AS and (south-)western Switzerland (see Stucki et al., 2012). On 3 October 1868, a similar distribution occurred with more (less) precipitation along the Alpine divide (northwestern Switzerland). On 4 October 1868, the precipitation rates were generally lower, and the highest daily precipitation was observed in (the southern parts of) Graubünden.



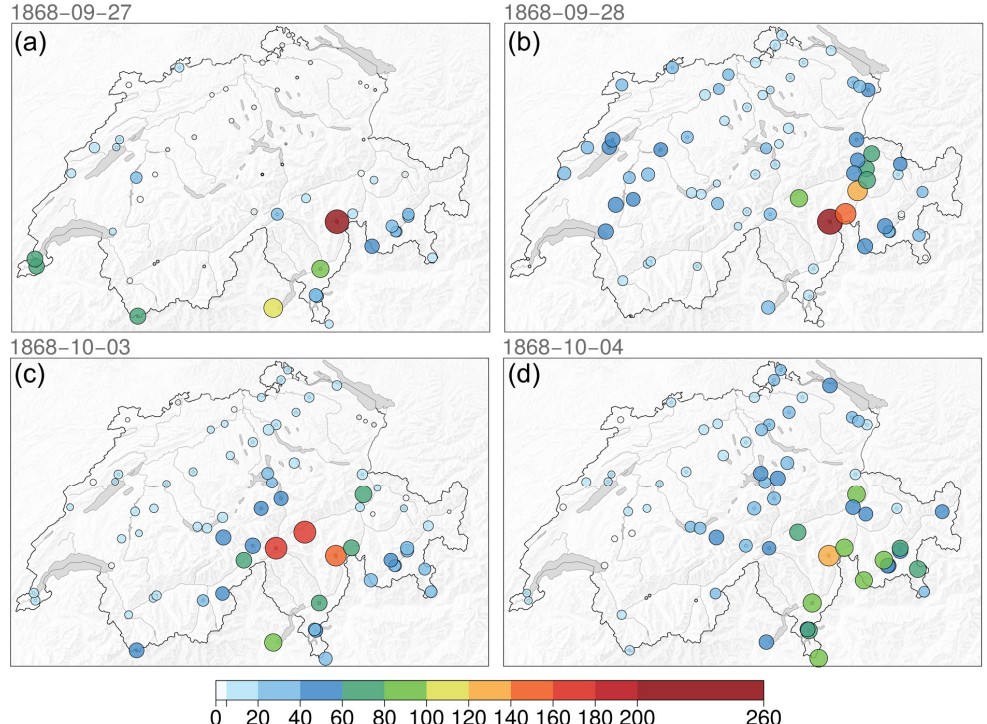

**Figure 5. Observations of daily precipitation (mm/24 hours from 05:40 local time) for (a) 27 September 1868, (b) 28 September 1868, (c) 3 October 1868, and (d) 4 October 1868 as collected from Wolf (1868), Coaz (1869), and Luino et al., (2005). Note that not all daily values are available for all stations.**

In summary, the spatial heterogeneity of precipitation indicates heavy thunderstorm activity during the episodes 09-27 and 10-03. The analyses show a local concentration of extreme precipitation over the AS that reached across the Alpine divide into (central) Graubünden for episodes 09-27 and 10-03. Very high precipitation amounts of up to 100 mm/24 h or more were also observed at stations located across the central Swiss Alps (Cantons of Valais and Central Switzerland), particularly during the 10-03 episode. The observed spatial distribution and intensities of heavy precipitation correspond well with the observed damage amounts in Fig. 2. As an exception, the damages reported for the western part of the Canton of Ticino cannot be verified due to unavailable precipitation measurements in this region.

### 3.3  Synoptic-scale atmospheric conditions

Here, we employ a PV- and IVT-perspective to look at the upper-level dynamics during all four episodes of heavy precipitation. The first episode (episode 09-17; see Sect. 3.2) was characterized by cyclonic Rossby wave breaking over Great Britain, subsequent amplification of the wave and development into a PV-streamer that reached northern Africa (on the 330-K isentrope) on 16 September 1868 (not shown). Until 17 September 1868, a downstream stationary ridge prevented eastward propagation of the system (Figure 6a). During this time, this



trough and a subsequent trough moving in from upstream merged into one broad trough associated with a very
strong moisture flux reaching the AS from the south-west (Figure 6c and e).

At the beginning of the second episode (09-22), another trough formed over western Europe on 21 September

4    1868 and cyclonical Rossby wave breaking occurred on 22 September 1868 (Figure 6b). The associated, very
strong south-westerly flow brought moist air towards the AS (Figure 6d and f). A strong ridge, located downstream
of the trough, persisted through 23 September 1868.

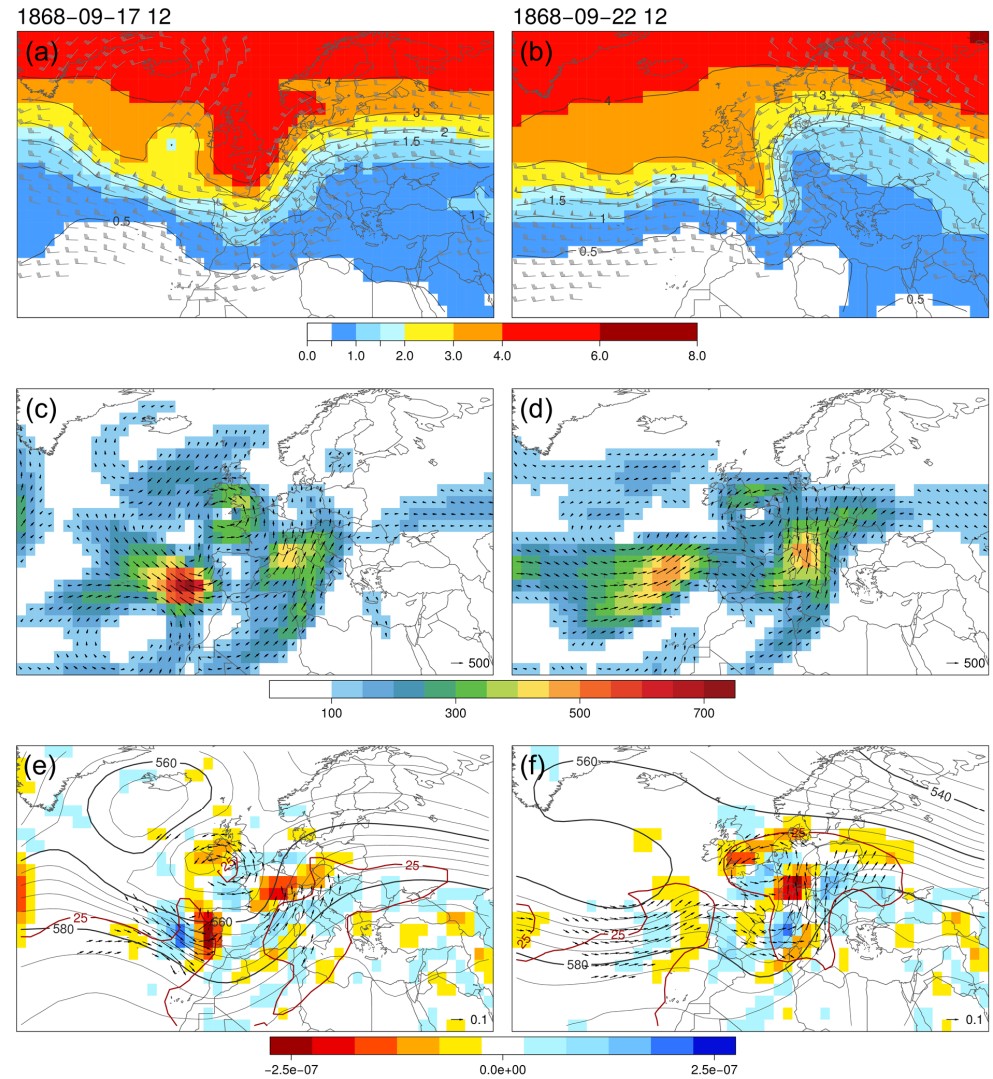

**Figure 6.** Synoptic analyses for 1868-09-17 12 UTC (left panels) and 1868-09-22 12 UTC (right panels). (a),
(b) PV units at the 330 Kelvin isentrope, and wind speed (in knots according to wind barbs) at 250 hPa. (c),
(d) Integrated water vapor transport (shade and vectors; kg m$^{-1}$ s$^{-1}$). (d), (e) Moisture flux divergence (g kg$^{-1}$
12   $^{1}$ s$^{-1}$; shade; negative values indicate flow convergence) and transport (m s$^{-1}$ kg kg$^{-1}$, vectors) at 850 hPa
levels, and precipitable water for the entire atmosphere (red contours indicate 25 kg m$^{-2}$).



The third episode (09-27) started with a subsequent trough over the central Atlantic that reached equator-ward to
30° N on 26 September 1868, an associated jet stream passing just north of Switzerland, and persistently strong

4  south-westerly moisture flux on the AS (IVT almost constantly 300 kg m$^{-1}$ s$^{-1}$) between 25 and 29 September 1868
(Figure 7c and e).  The trough then moved slowly eastward and underwent anticyclonic wave breaking on 28
October 1868 (Figure 7a).

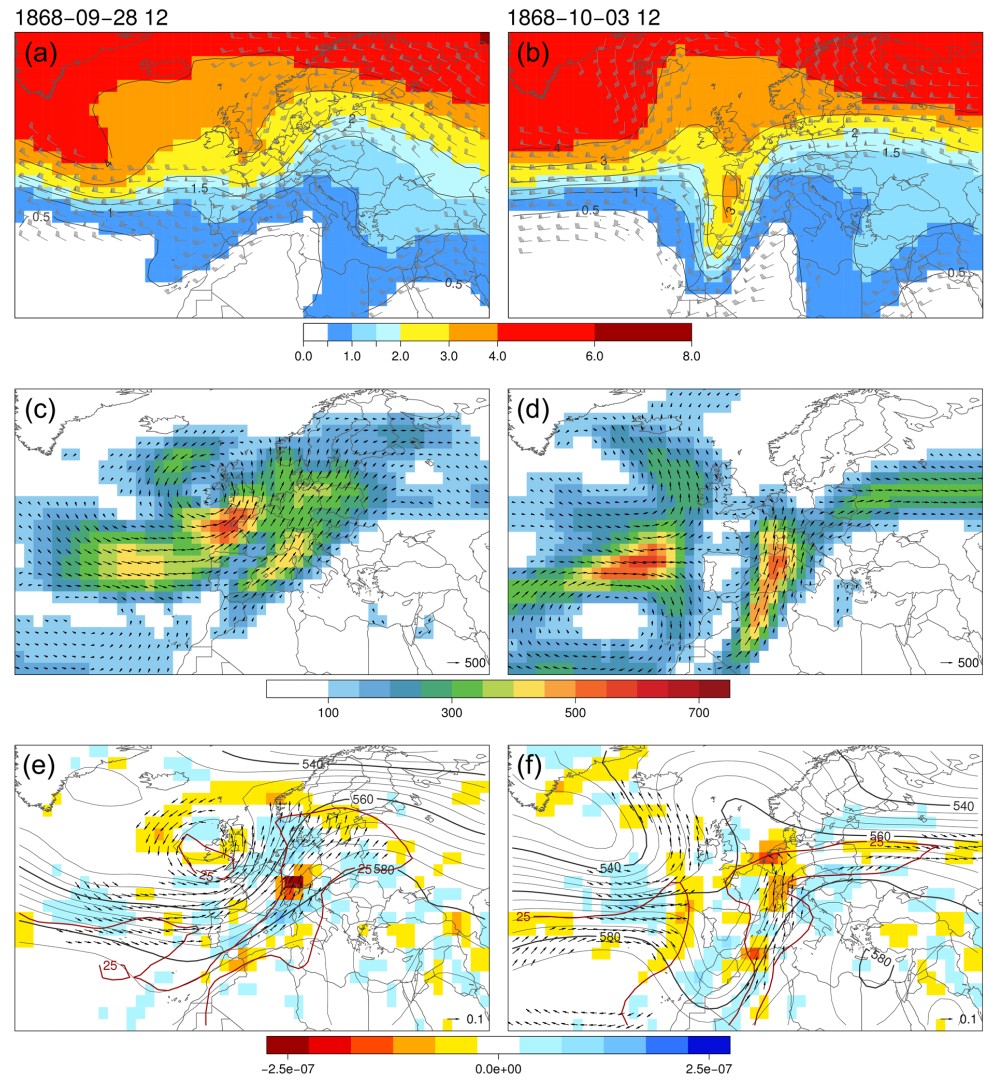

**Figure 7. As in Figure 6, but for 1868-09-28 12 UTC (left panels) and 1868-10-03 12 UTC (right panels).**





A next, meridionally extended low-PV trough reached western Europe on 30 September, remained stationary due to a strong ridge downstream (not shown), then amplified and turned into a PV-streamer over the next days (the fourth episode; 10-03; Figure 7b) until its decay on 4 October. The episode was accompanied by strong and

4  persistent moisture flux from the southwest (IVT >300 kg m$^{-1}$ s$^{-1}$) to the AS (Figure 7d and f), but shifted slightly from south-westerly to southerly this time. Maximum IVT values on the southwestern AS were reached on 3 October 1868.

In summary, the synoptic-scale analyses reproduce the well-known features of a strong, elongated, and meridional

8  PV-streamer just west of the Alps. Strikingly, the large-scale atmospheric conditions during the antecedent episodes 09-17 and 09-22 evolved in parallel to the episodes 09-27 and 10-03 (Figure 6; Figure 7) with only small differences: On 17 September 1868, the moisture flux was slightly more southerly compared to 28 September 1868, and the center of convergence was further to the west. On 22 September 1868, the PVU contours and the

12  500-hPa isolines show potential cyclonical wave breaking just west of the Alps, while breaking is more uncertain for 3 October 1868.

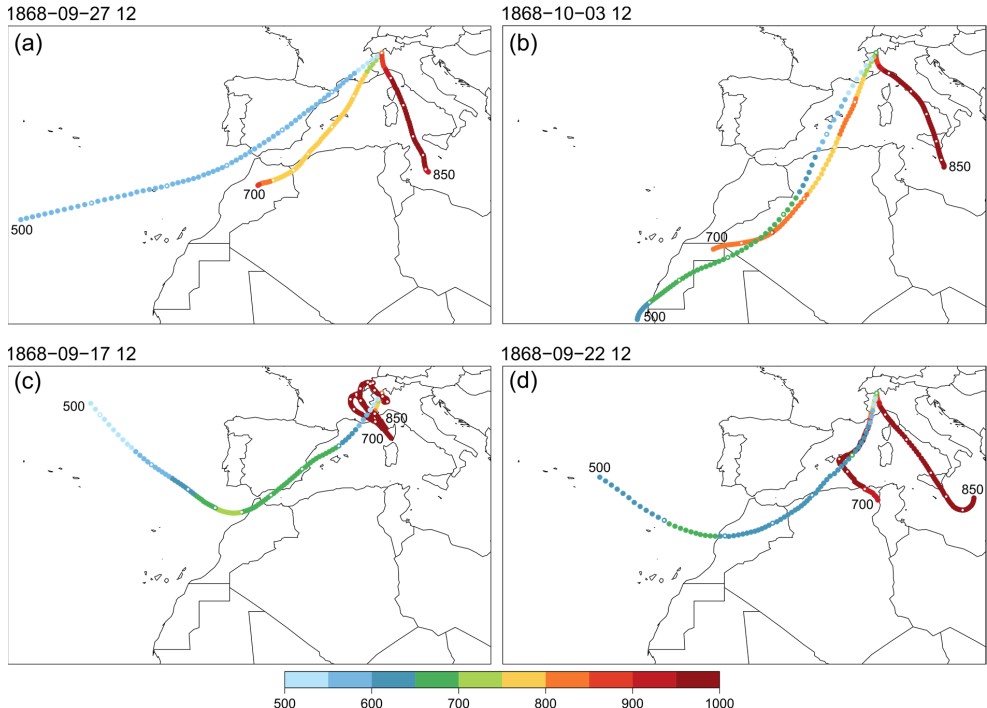

16  **Figure 8. Backward trajectories of 3 days (5 days for panel (c)) ending at 850 hPa, 700 hPa, and 500 hPa levels over the grid point 46° E 8° N in 20CR on (a) 27 September 1868 12 UTC, (b) 3 October 1868 12 UTC, (c) 17 September 1868 12 UTC, and (d) 22 September 1868 12 UTC. Color segments indicate pressure levels in hPa along the trajectories. Time steps of 12 h are indicated by white dots. The 20CR data are provided**
20  **by the NOAA-ESRL Physical Sciences Division, Boulder, Colorado via their web site at http://www.esrl.noaa.gov/psd/, and produced using the parcel trajectory software (traj3d) available from the University of Melbourne at http://www.cycstats.org/trajectories/trajhome.htm.**



Backward trajectories from a 20CR grid point over the Alpine divide just west of the LMR (at 46° E 8° N) further illustrate the almost parallel development of the four episodes. They indicate that air parcels ending on the respective key days of all four episodes (Figure 8) took very similar paths over time scales of a few days before

4 arrival. In all four episodes, air parcels arriving at 850 hPa (700 hPa and 500 hPa) had propagated north-westward (north-eastward) from the central (western) Mediterranean Sea, and only episode 09-17 is less consistent due to slow propagation at lower levels. Over all levels, advection seems more tilted towards sector south-west (southeast) for the episodes 09-17 and 09-27 (the episodes 09-22 and 10-03).

8 In conclusion, the repeated occurrence of flood-inducing atmospheric patterns over almost three weeks set the synoptic-scale atmospheric conditions for the record water level of Lago Maggiore and a devastating flood event. The upper-level dynamics over the North Atlantic during late September and early October 1868 as portrayed in 20CR offer plausible explanations for the formation of such a series of heavy precipitation episodes on the AS.

12 They are in very good agreement with similar, more recent cases such as the severe flood on Lago Maggiore in September and October 2000 (e.g., Froidevaux and Martius, 2016). During this recent event, several episodes of heavy precipitation occurred in close succession (Barton et al., 2016). The final episode lasted for several days with a downstream blocking anticyclone over central and eastern Europe, preventing an eastward progression of

16 the breaking wave / PV-streamer (Lenggenhager et al., in prep.).

Even so, several recurring tropical cyclones in the Atlantic were crucial for the meridional amplification of the mid-latitude flow during the 2000 case, and hence the formation of PV-streamers over western Europe. However, no such cyclone is visible in 20CR for the 1868 case. A cyclone may not have existed, or this absence points to

20 potential flaws in 20CR over the western (sub-) tropical Atlantic for this period, at least in the ensemble mean (cf. Brönnimann et al., 2013; DiNapoli and Misra, 2012). In addition, some of the troughs over the Atlantic tend to have decreasing phases in 20CR between the precipitation episodes (not shown), whereas one would expect a continuous intensification and then breaking.

24 **3.4    Meso-scale atmospheric conditions**

Given the coarsely resolved 20CR dataset, the analyses above also cannot explain local phenomena like the observed patches of precipitation in the LMR and across the Alpine divide. For this, we look at a number of simulated atmospheric variables on local to regional scales, i.e. from the innermost WRF domain (see Sect. 2.2).

28 A first comparison of simulated and observed precipitation rates (Figure 4) at six rain gauge stations shows generally lower absolute rates in the simulation. Nevertheless, the observed temporal evolution is well reflected, as is the topographic repartition of extreme amounts at the high-elevation mountain passes and lower values in the lowlands.

32 To expand the spatial perspective, we compare maps of simulated precipitation sums for 28 September and 3 October 1868 (Figure 9a and b) with the observed precipitation sums at measurement stations for the same days (Figure 5b and c). For both days, the simulated and observed precipitation corresponds well and the simulation correctly places distinct maximum intensities on mountain ridges and tops in or close to the LMR, i.e. in the

36 vicinity of the observed maxima. The main area of precipitation is simulated as a southwest- /northeast-oriented narrow triangle that starts just south-west of the LMR. In the simulation, this triangle is more elongated on 3




October than on 28 September 1868 and reaches further across the Alpine divide towards the western, northern and eastern parts of Switzerland.

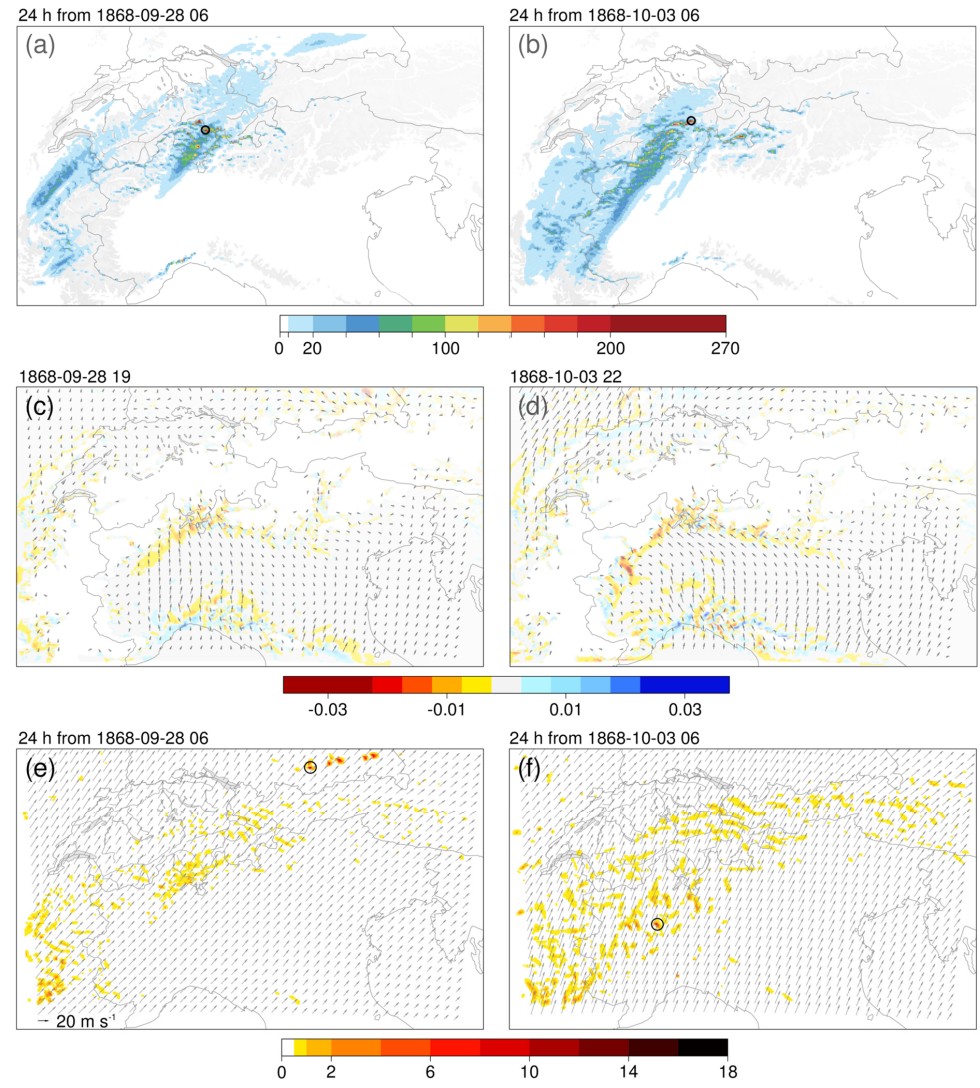

**Figure 9. Weather variables for 1868-09-28 (left panels) and 1868-10-03 (right panels) as simulated in the innermost WRF model domain. (a) and (b) Cumulated precipitation (mm/24 hours) starting at 06 UTC. The color scale is as in . Orography is shown above 1000 m a.s.l. (c) and (d) Moisture flux divergence (g kg$^{-1}$ s$^{-1}$; shade; negative values indicate flow convergence) and transport (m s$^{-1}$ kg kg$^{-1}$; vectors) at 900 hPa at two specific time steps. (e) and (f) Mean vertical wind (uplift w; m s$^{-1}$) and mean horizontal wind speed at 600 hPa over 24 h starting at 06 UTC. Circles mark the grid cells with the highest values.**



While the simulation works very well in the core area of precipitation on the AS, it performs less well in other areas. For instance, the very high amounts of precipitation observed in central Graubünden are not fully simulated for 28 September 1868. For 3 October, some of the secondary regions of intense precipitation are well simulated

4 (e.g. the scattered precipitation over north-western Switzerland, over central Switzerland, and over Graubünden), but mostly underestimated. This indicates that in reality, intense precipitation has arguably reached further into northern Switzerland.

Overall, the simulated precipitation not only corresponds well with observations, but also with the damage reported

8 in the documented areas of the AS (Figure 2). Although there are no observations in the south-western part of the precipitation triangles, we infer from this very good correspondence that the high damage amounts documented in this region were directly linked to high precipitation rates. In addition, the WRF model is able to reproduce the observed intensities on the AS, just not at the exact grid point or with the exact timing. Despite some

12 underestimation of precipitation on the other side of the Alpine divide, we infer that the ingredients and processes necessary to form the episodes of heavy precipitation are represented in the WRF model, at least for the AS. In turn, this inspires confidence for a more detailed formulation of the flow modulation in this region.

While the reanalysis data show large-scale, strong moisture flux convergence over and across the Alps (Figure 6;

16 Figure 7), the downscaled data show small-scale patterns of both convergence and divergence near the surface. As an example, Fig. 9c and d show two instances of moisture flux over the Po plain (south of the masked Alpine orography) towards the concave topography on the AS. In the LMR, moisture is primarily advected from easterly to south-easterly directions. During both episodes, there is strong low-level moisture convergence along the

20 foothills west, north, and east of the Lago Maggiore, with a slightly enhanced easterly (westerly) component during the first (second) episode. Winds veer to more southerly and southwesterly directions in the mid-troposphere (600 hPa level; Figure 9e and f). The mean flow on 28 September 1868 is clearly southwesterly over the Alps, while it has a more southerly component on 3 October 1868. In addition, patches of strong vertical uplift are simulated in

24 the triangle-shaped area of heavy precipitation. Obviously, the WRF simulation is able to reflect the production of convective cells upstream of the strongest precipitation intensities.

These dynamics are consistent with the observed precipitation patterns in Switzerland (Figure 5), as well as with recent high-precipitation events, where the precipitation patterns were found to be related to moisture flux that

28 reaches western Ticino and eastern Graubünden from sector SW and the Gotthard region from sector S (Froidevaux and Martius, 2016). Furthermore, they are consistent with the orientation of the upper-level trough (Figure 7; see also Stucki et al., 2012): Heavy precipitation concentrated in the western (eastern) LMR is prevalent with PV-streamers that adopt a south-east / north-west (south-west / north-east) tilt (Martius et al., 2006).

32 To further investigate the vertical structure of the simulated troposphere, two WRF model soundings are taken at the location of today's Milano Linate Airport (LIML; Figure 10), the first for 27 September 1868 12 UTC and the second for 3 October 1868 12 UTC. The soundings show similarities with real soundings taken at the same location during flood-producing weather situations in the LMR, e.g. during the 24 September 1993 and 14 October 2000

36 floods (not shown here; see Panziera et al., 2015; and http://weather.uwyo.edu/upperair/sounding.html). Both soundings reproduce the veering of the wind with height, that is, from easterly directions within the planetary





boundary layer to south-westerly in the mid-troposphere. In addition, two observations at Milan and Mantua confirm low-level easterly winds over the Po plain (Maugeri et al., 1998).

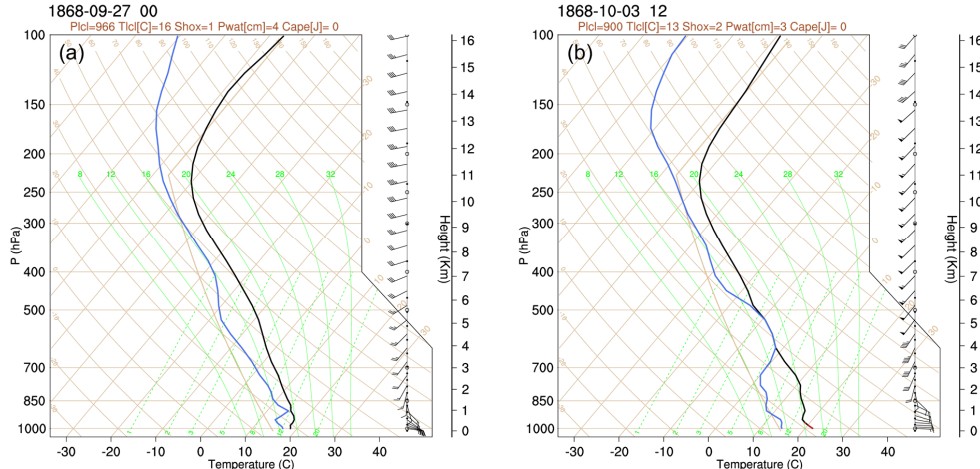

**Figure 10. Simulated sounding from WRF model output at the Milano Linate Airport (LIML; see Fig. 11b for location) for (a) 27 September 1868 12 UTC and (b) 03 October 1868 12 UTC.**

8    In summary, the wind velocities, low-level convergence, mid-level uplift and northeastward transport shown in Fig. 9, and the two model soundings are in line with the low-level jet concept by Panziera et al. (2015; their figure 21). The higher wind velocities of the 03-10 sounding compared to those of the 09-27 sounding, especially at high altitudes, are a plausible cause for the wider spatial extension of the heavy rainfall reaching into the inner Alpine

12   regions (Figure 9; see also Fig. 5). Further similarities to Panziera et al. (2015) include the relatively large spread between surface temperature (around 23 °C) and dewpoint temperature (approx. 17–18 °C), the lifting condensation level at approximately 900 hPa, and fully saturated, conditionally unstable to neutral air layers between 3.5 and 5.5 km above sea level. A significant low-level spread (between ambient and dew point

16   temperature) is typical for a Po valley sounding, even during heavy precipitation events on the AS. Nevertheless, the spread seems to be too large on 27 September 1868 in the model sounding compared to the real soundings during the 1993 and 2000 cases. CAPE is rather low (approx. 200–300 J; maximum approx. 800 J during the episodes 09-27 and 10-03) most of the time. In nearly moist-neutral conditions, deep convection can develop in a

20   low-CAPE environment, especially if forced orographic lifting helps to initiate updrafts (cf. Panziera et al., 2015; especially their figure 15).

Such a scenario of persistent precipitation, with intermittent occurrences of thunderstorms, is compatible with the heterogeneous precipitation patterns in both simulations and observations. It is also in good agreement with eye-

24   witness reports by Coaz (1869) and observer remarks in Wolf (1868). For the episode 09-27, both authors reported a core area of very intense and persistent thunderstorms along the Alpine divide near San Bernardino, accompanied by hail, while thunderstorms were arguably less severe in other places, e.g. towards the western parts of Graubünden / Central Switzerland. For the episode 10-03, very high thunderstorm activity was again reported,



even combined with hail in some places, and they inferred a larger thunderstorm-affected area than in the previous episode.

In summary, the meso-scale simulations are able to reproduce the main characteristics described by Panziera et al.

4 (2015) regarding the structure and dynamics of elongated precipitation bands that are triggered in or south-west of the LMR and stretch well across the Alpine rim, and that can be maintained over many hours to a couple of days. More generally, the meso-scale flow patterns agree well with findings from the Mesoscale Alpine Programme (relative to the convective IOP2b case; see Rotunno and Houze, 2007), where south-easterly low-level winds in

8 the Po Valley, together with synoptic-scale southerlies at higher levels, lead to enhanced confluence in the LMR and thus increase precipitation intensity in this region.

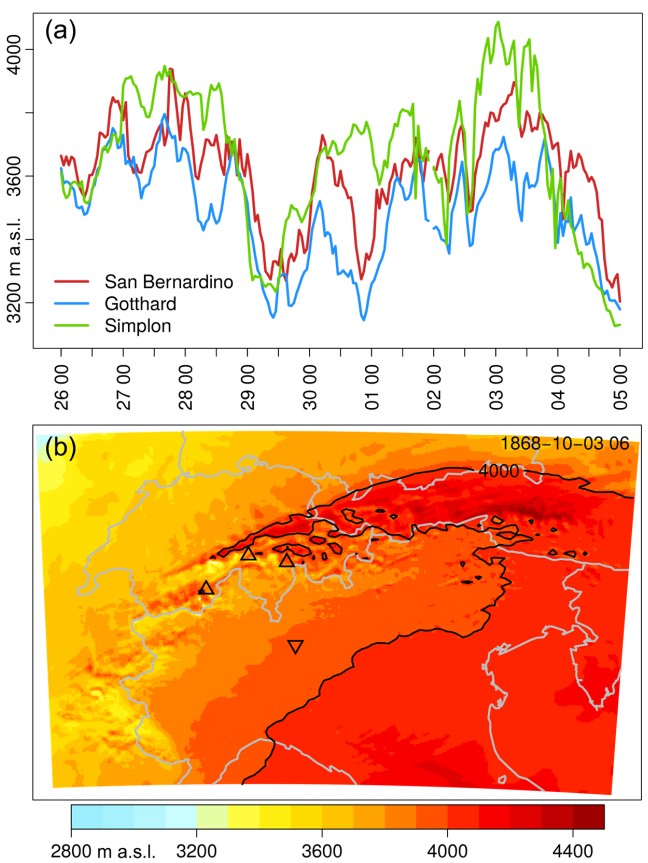

12 **Figure 11. (a) Freezing level at San Bernardino, Gotthard, and Simplon passes between 26 September 1868 00 UTC and 5 October 1868 00 UTC (x-axis; day of the month and hours UTC are given), as calculated for the closest grid point in the WRF simulation. (b) Height of the freezing level (shade and contour at 4000 m a.s.l.) for 1868-10-03 06 UTC. Small white spots indicate orography reaching above the freezing level height.**

16 **Upside triangles indicate the locations of the San Bernardino, Gotthard, and Simplon passes from east to west; the downside triangle indicates the location of the Milano Linate Airport.**





Besides precipitation intensity, the associated freezing level is a decisive factor for runoff formation. For instance, Bundesamt für Wasser und Geologie (2002) and Grebner (1993, 2000) found a freezing level above 3500 m a.s.l. for the 1993 flood, and a freezing level that was persistently above 3000 m a.s.l. during the 2000 flood. In our simulation, the freezing level in the atmospheric column over LIML reaches approximately 3800 m a.s.l. on 27 September 1868 12 UTC and 3 October 1868 12 UTC, and reaches up to >4000 m. a.s.l. over the Alpine divide during these two episodes (Figure 10; Figure 11).

In the WRF model, substantial growth of the snow layer is only simulated for the Monte Rosa range in the westernmost tip of the Lago Maggiore catchment (daily accumulated snow between 12 cm on 27 September 1868 and 85 cm on 4 October 1868; not shown). This means that almost all precipitation in the LMR fell as liquid water, with hardly any snow storage that would delay runoff. It is only between these episodes that the freezing level drops towards 3200 m a.s.l. in the simulation. This is again in good agreement with Coaz (1869): Reportedly, the daily maximum temperature at San Bernardino Pass (2100 m a.s.l.; on the Alpine Divide) was 8.2 °C on 27 September 1868. On the same day, the temperature was 13.5 °C at Splügen (1500 m a.s.l.; cf. 16 °C in Petrascheck, 1989). It was still 4 °C on 28 September 1868 on San Bernardino pass, and snow was only reported near the mountain tops, (i.e., >3000 m. a.s.l.) on 29 September. On 3 October 1868, the temperature was 14 °C on San Bernardino Pass, and 18 °C at Splügen (Petrascheck, 1989). Assuming (nearly) saturated air layers and moist-adiabatic lapse rates in the region, these temperature observations point to freezing levels near or even above 4000 m. a.s.l. (cf. Stucki et al., 2012). It is only on 4 October 1868 that snow was reported for the high-elevation stations (e.g. 5 cm at Grand St. Bernard).

### 3.5  Hydrological response: Simulated runoff and lake levels

The hydrological simulations for the period between October 1867 and November 1868 are driven by the meteorological fields derived from an analog resampling method as a base scenario (Figure 12). Two further scenarios are nested in this simulation: one comprising the higher level of detail from the WRF downscaling simulation over the full available period of ten days (26 September to 5 October 1868), and one that reflects the deforestation present in 1868, with 50 % less forest cover compared to today. The simulations show that the flood event followed a rather dry to normal late-summer period, and a first very wet spell during September. During this wet period, soil storage capacity was reached and the Lago Maggiore lake level rose significantly. For the second wet period (comprising the episodes 09-27 and 10-03), WRF simulations are added to the analog-driven simulations. Differences appear between the analog- and the WRF-derived precipitation: WRF shows a decreasing intensity for the four major rainfall events, while the analog method shows increasing intensities (Figure 12, left panel). Total precipitation sums are approximately equal for both approaches. The simulated lake level responds accordingly (Figure 12, right panel), i.e. with a significant increase in the lake level during the early treatment period in the WRF simulation, and a smaller increase later (and vice versa for the analog method).

Although our hydro-meteorological model chain tends to overestimate the lake levels (see Sect. 2), we are not able to reproduce the observed lake level peak. Possible reasons are manifold. Firstly, the reconstructed precipitation input could have been strongly underestimated. However, given the very steep rise in the observed lake level, this precipitation amount would have been enormous. Secondly, a damming of the Lake due to a log jam could also have occurred. We evaluated an idealized model experiment of such a log jam by reducing the outflow amount to one third of the theoretical relationship for the time of the steepest increase. At the time of the highest peak, such





a log jam at the Sesto Calende bridge would have needed to be resolved in one day to match the steep decline observed in the lake level records. However, to our knowledge, such a log jam is unreported in the historical documents. Thirdly and most likely, the reason for the strong underestimation is due to changes in the lake level–

4    outflow relationship. Ambrosetti (1994) reviewed long-term lake level changes in the 19[th] and 20[th] centuries and stated that the high lake levels recorded before 1868 are unlikely to be achieved again, as a ridge on the lake outlet was heavily eroded during the 1868 flood event (minus 30 cm). As we do not have any hint of the shape of this ridge and the cross-section at the lake outflow, we cannot prove this theory. However, we assume that changes in

8    the lake level–outflow relationship is the most likely reason, among the ones discussed.

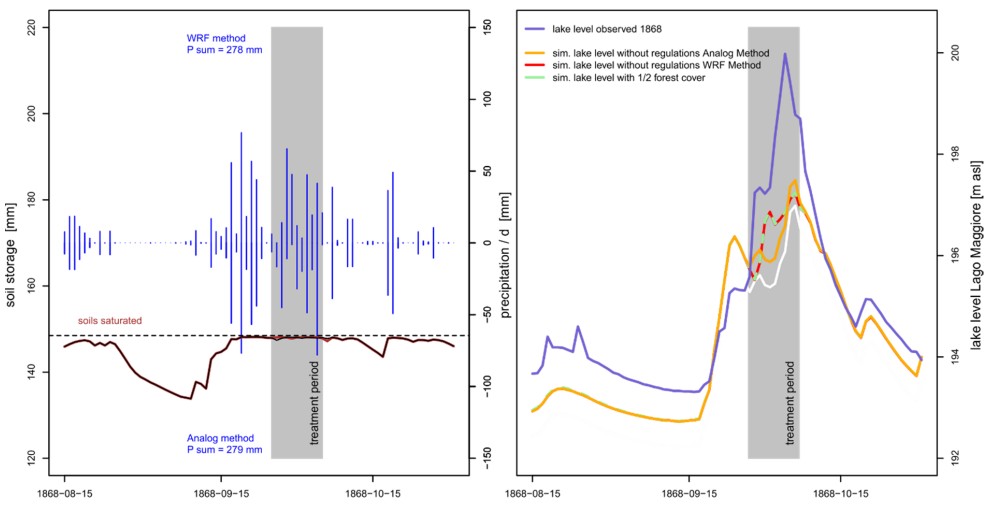

12    **Figure 12. Hydrological simulation of the 1868 flood event in the LMR. The left panel displays the precipitation input to the system (area averages in mm) for the WRF downscaling simulation (positive values) and the analog resampling method (negative values), as well as mean areal soil storage. The right panel shows the observed (blue lines) and simulated (colored lines) lake levels from August to October 1868.**

16    **WRF scenarios are only simulated for the treatment period shaded in grey. The deforestation scenario was computed for a full hydrological year from 1 October 1867 to 31 October 1868.**

Furthermore, we estimate the effect of deforestation on the flood event in 1868. Our simulations show only very

20    small differences, indicating that even a higher percentage of forest cover (conditions similar to today) would not have dampened the impacts of such an extreme event. This is mainly due to the limited and rapidly filled storage capacity of the vegetation, and especially of the mostly very shallow soils (Figure 12, left panel). This is confirmed by literature values regarding forest canopy and forest floor interception capacity, which are about 3 mm at

24    maximum for beech (*fagus sylvatica*; Gerrits et al., 2010) and 4.7 mm at maximum for spruce (*picea abies*; Breuer et al., 2003). In relation to a total precipitation input of 280 mm, the limited significance of forest cover during such extreme events is clear.





## 4        Summary and conclusions

In this study, we have combined traditional reconstructions and numerical simulations to assess an extreme flood that occurred in the Lago Maggiore catchment and surrounding areas on the south side of the central Alps during

4        the end of September and early October 1868. We have taken an interdisciplinary approach to describe damage and impacts, the flood-inducing precipitation, the driving atmospheric conditions, and potential reasons for the extreme flood levels.

The analyses show that the systematic use of contemporary sources, such as documentaries, weather annals, and

8        eye-witness and administrative reports, results in a very detailed picture of the impacts and damages on regional and even municipality levels. Damage was concentrated around Lago Maggiore and south of the Alpine divide, but also reached across the Alpine divide into Graubünden, Valais, and Central Switzerland.

The synoptic (large-scale) atmospheric conditions have been analyzed using 20CR, a global reanalysis dataset.

12      The synoptic situation is comparable to modern cases. However, we additionally find a repeated occurrence of similar patterns over four distinct episodes from 17 September to 4 October 1868. Typical precursors of heavy precipitation are found with PV-streamers, atmospheric wave breaking, and persistent, very high (IVT >300 kg m$^{-1}$ s$^{-1}$) southerly moisture fluxes towards the Alps.

16      Simulated weather variables at meso- (local) scales are obtained from dynamical downscaling 20CR input to a 2-km grid using the limited-area weather model WRF. The simulated precipitation rates, patterns, and atmospheric dynamics agree very well with the observed precipitation and damages. They also agree well with existing concepts of forced moisture convergence in the concave orography around Lago Maggiore, and subsequent uplift,

20      convection and repeated northeastward propagation of convective cells. Hence, the WRF model is able to reproduce small-scale phenomena of an event that occurred in 1868, and to achieve a realistic cloud-resolving simulation with downscaling from 20CR.

In addition, we simulate freezing levels that are mostly close to 4000 m a.s.l., meaning that virtually none of the

24      heavy precipitation was stored as snow and delayed runoff. The subsequent hydrological modelling captures the gradual increase in the Lago Maggiore water level. The simulated and observed peak levels differ by approximately 2 m. Reasons for this gap may be found in unknown riverbed profiles at the lake outflow. Reduced forest cover in the Lago Maggiore catchment at that time did not have an influence on the extreme response of Lago Maggiore;

28      experiments with different percentages of forest cover have not resulted in different responses in the lake level.

This is in contrast to leading forestry policies at that time, which presumed a forest storage effect. Based on studies of French engineers from the late 18th century, the idea of a general correlation between forest cover and floods was established in Switzerland in the mid-19th century and strongly supported by the Swiss forestry association.

32      The catastrophic flood in 1868 triggered the implementation of afforestation programs and the construction of hydraulic structures. Following the event, the federal government enacted forest and hydraulic engineering laws, and committed to financially supporting protection measures. As a result, many rivers were channeled, torrent controls established, and alpine meadows afforested in the following decades. A differentiated consideration of

36      the role of forest cover as a flood protection measure has only taken place since the late 20th century, and the paradigm has only recently shifted towards re-naturalization of waterways.

In conclusion, we find – for this specific case – very good agreement between traditional reconstructions and numerical simulations on almost all levels, mind you for an event in the mid-19th century. To our knowledge, the





provided local-scale weather maps are the earliest to date, but we hope that this is not the case for long. Indeed, these findings highlight the broad prospects for these combined analyses with respect to studies of weather events and their impacts back to the early 19th century, and maybe beyond. They also highlight the opportunities for cross-validations between scientific disciplines, be it meticulously examined documents for validating weather or runoff models, or numerical support for historical climatology studies.

**Acknowledgements**

The work was supported by the Oeschger Centre for Climate Change Research and by the Swiss National Science Foundation (project CHIMES). We would like to thank Ralph Rickli, Christian Rohr and Martin Stuber for fruitful discussions about weather dynamics and historical aspects. We also thank Erin Gleeson for the careful language editing. Support for the Twentieth Century Reanalysis Project version 2c dataset is provided by the U.S. Department of Energy, Office of Science Biological and Environmental Research (BER) program, and by the National Oceanic and Atmospheric Administration Climate Program Office.

**Appendix**

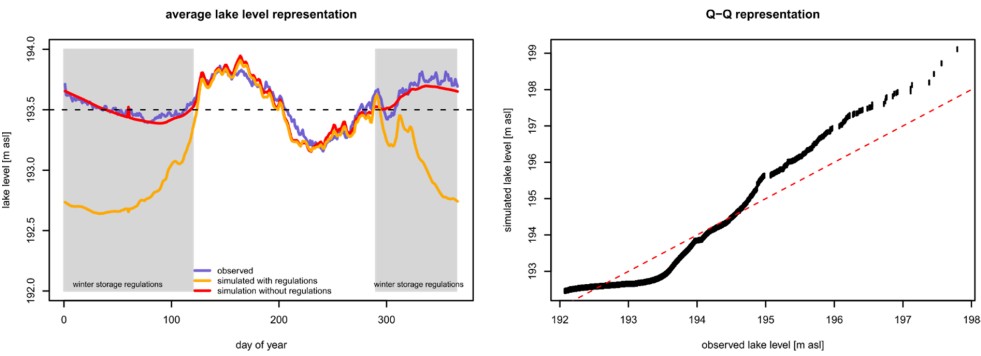

**Figure A1: Validation of the hydrological model performance by means of long term average (1960–2010, left panel) and daily value-based Q-Q representation, indicating a fair model performance with the tendency to overestimate higher lake levels. In the left panel, we show the model representation assuming a simple winter storage regulation (red line).**

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
