# Peer review of "Reconstruction and simulation of an extreme flood event in the Lago Maggiore catchment in 1868"

_Natural Hazards and Earth System Sciences, 2018_

## Referee Comment (RC1) · Anonymous Referee #1 · 11 Jun 2018

**Review of the paper "Reconstruction and simulation of an extreme flood event in the Lago Maggiore catchment in 1868", by P. Stucki et al.**

**General comments**

This paper describes a disastrous hydro-meteorological event that affected Switzerland 150 years ago. Thanks to the availability of the "Twentieth Century Reanalysis" dataset, meteorological and hydrological state-of-the-art models have been applied, to try to obtain quantitative results that complement the documentary/historical data available.

The approach is interesting, and this is probably the earliest event that has been reconstructed/simulated with models, comparing the model results with the earlier observations. The paper is well written and, in my opinion, basically it is worth of publication in NESS. However, I am a bit concerned with the fact that the reanalysis at that time is based on very scarce data (essentially sporadic surface observations) if compared to nowadays situation, so that it is affected by large uncertainties (particularly at small scales and in atmospheric moisture content) that may adversely reflect on the simulations, especially those at high resolution. The latter can be useful since they provide more accurate description of the orography and land properties and of the atmospheric dynamics, but one cannot rely too much upon detailed results of the model simulations: this limited predictability problem at the convective scale severely affects the quantitative precipitation forecasts even in our days. I can only imagine how severe it is at the time of the event! Therefore, I think that the authors should devote more space to try to quantify or at least describe qualitatively if not quantitatively the uncertainties of their results.

**Punctual comments**

The Introduction appears a bit too long and detailed – it contains some treatment that should be postponed to the specific chapters.

Page 1, line 36: if AS stands for Central Alps, it should be CA. But in the context of the paper I had the suspect that it means "Southern Alps" (or Alps South?). I suggest, in any case, to use initials that correspond to the English terms.

Page 2, lines 1-7: the paper Malguzzi et al (2006) (see also below "References") should be referenced here, with some short description, because it is very similar to the present paper in various aspects: it describes a major historical (1966) flood event that affected the (Eastern) Alps (and not only – it was considered the century flood in Italy). It applies a very similar approach (meteorological and hydrological model chain), although the used reanalysis (and also the verification) takes advantage from a much better data coverage than available one century before!

Page 2, line 9: LMR: Lago Maggiore Region?

Pages 2-3: somewhere in the Introduction (and not only at page 12), the MAP international project must be mentioned (for example quoting Bougeault et al, 2001): its major objective was to study (from the observational and modelling point of view) the atmospheric processes related to heavy precipitation and flood the Alps, with its largest observational effort concentrated just in the Lago Maggiore area.

Page 3, lines 5-18: regarding the PV streamers west of the Alps, I think that the main interesting aspect (related to the orographic forcing) is that the (orographic) precipitation occurs more to the east, with respect to the position of the PV anomaly, than expected in the case of flat terrain. In other terms and using more traditional synoptic concepts, while in the flat case precipitation has to be expected ahead of the cold front, more or less in the area of the warm conveyor belt, in the case of the Alps (or similar orography perpendicular to the more-or-less southerly flow) precipitation may be heavy also in the warm sector well in advance of the cold front. It must be remembered, in any case, that the orography can change the synoptic scale flow at scales larger than those of the orography itself.

Page 4, lines 19-25: please give some more info here about the 20CR for those not familiar, in particular for what concerns the input observations (other variables besides surface pressure? T and humidity?), the available variables on the pressure surfaces and at the ground) and the degree of uncertainty as estimated in the literature.

Page 5, lines 5-8: the application of the nudging, although justified by the need of constraining the forward meteorological model to run close to the reanalysis "trajectory", has drawbacks that should be mentioned: for example, this is not a real "hindcast" experiment – how the precipitation forecast differs from a pure forward integration, i.e. without nudging? (one test should be made at least for one case).

Page 5, lines 21-25: please explain better the procedure – include some words about the quality/limitations/uncertainties of the E-OBS dataset.

Page 6, lines 15-17: there is similarity with the 1966 event described in Malguzzi et al: in both cases there is strong enough precipitation at the divide and on the downstream side of the Alps to cause flooding also of the rivers flowing on the north side. It is not clear, however, to what extent such precipitation "originates" on the upstream side (due to transport of cloud condensates and hydrometeors across the Alpine crest) or on the downstream side (including possible thunderstorms). Model results (see Fig. 8) do not seem to represent this aspect.

Page 6, line 24: it is not clear if only the "inflation" of the Swiss Franc is considered here or also the fact that the nowadays (economic) damages would be much larger due to the real value and vulnerability of today's infrastructures, resources etc. It seems to me the first is the case here, but the second would be more interesting...

Page 8, lines 11-14: I think that the most dramatic aspect, which makes this event so exceptional, is the occurrence of an entire sequence of heavy precipitation cases. Of course, this aspect cannot be "explained" in terms of atmospheric dynamics, even with the help of model simulations.

Page 8, line 37 (and elsewhere, for ex. page 9, lines 25-26): the expression "Rossby wave breaking" is used several times, but I think that not only it is too technical, it is also ambiguous and should deserve an explanation in more traditional terms.

Page 9, lines 2 and 15; page 10 line 1: a blocking anticyclone (in the traditional meteorological literature) is something different (in brief, an anticyclone located at lat. 50-70 over the Atlantic, Pacific or northern Europe, deviating the westerlies for several days). A ridge downstream of a trough is (I believe) a component of the same Rossby wave, that can become almost stationary. So I think that it not very correct to say that the ridge blocks the easterly propagation of the system.

Page 9, line 14: I guess "high-PV", not "low-PV".

Page 9 line 17: perhaps an "atmospheric river"?

Page 9, lines 27-33: qualitatively similar results have been obtained from trajectory computations by Bertò et al, 2004 (among others), for similar events of heavy precipitation on the southern side of the Alps. I suggest to quote this paper (see below the full reference).

Page 9, lines 36-37: I do not think there is a "plausible explanation" – after all, it is not given here.

Page 10, lines 5-8: some more clear should be provided here (or above) about the uncertainties of 20CR.

Section 3.4 is a bit too speculative, given the uncertainties as mentioned above (see the major comments). If uncertainties affect the low resolution, they can only be larger at higher resolution.

Page 10, lines 31 and 33: "well2 and "very good" seem too optimistic, at least in absolute terms (for both amount and spatial distribution of precipitation). I agree that the simulation provide useful meteorological information, perhaps better than expected, but I would use the word "satisfactory" rather than "very good".

Page 11, lines 7-9: these aspects were investigated during the MAP field experiment (as mentioned below). Here I would quote the paper by Rotunno and Ferretti (2001), providing an interesting explanation of the enhanced convergence due to horizontal inhomogeneities of moisture content at low levels in the incident flow.

Page 11, line 14: quite typical when convective scale are treated explicitly!

Page 11, line 35: wouldn't it be simpler saying "relatively low relative humidity".

Page 12, line 32: drop "were".

Page 13, regarding the effects of forests: I agree that forests may not mitigate flooding, however I wonder (but I am not an expert!) if they may act to reduce landslides (a very important cause of casualties and damages - see page 6, line 25) and reduce the amount of mud in flooded water that may worsen the effects.

Page 14, line 8: I think that the moisture flux is not a "precursor" of precipitation like the other factors, since it is almost contemporary to precipitation.

Page 14, line 12: "very well": again too optimistic and not consistent with the results presented here.

**Figures**

Fig. 4: the simulated precipitation seems rather low compared to the observed one. Note also that the Simplon station used to sample the simulation is not included in the station list for the observations – why?

Fig. 6 and 7 are not very clear. This is due mainly to the discontinuous (boxes) coloured fields (shade). This should be avoided using an interpolating graphical package.

Fig. 7: panel d) suggests the presence of an "atmospheric river".

Fig. 9a and 9b: it is very difficult to appreciate the differences/similarities with the observations of fig. 5. Perhaps the above figures should be enlarged, plotting on them the observations of fig. 5 with circles.

**References**

Page 18, line 23: Lago Maggiore (capital).

Suggested additional references:

Bertò, A., A. Buzzi and D. Zardi, 2004: Back-tracking water vapour contributing to precipitation events over Trentino: a case study. Meteorol. Z., 13, 189-200.

Malguzzi, P., G. Grossi, A. Buzzi, R. Ranzi, and R. Buizza, 2006, The 1966 'century' flood in Italy: A meteorological and hydrological revisitation. J. Geophys. Res., 111, D24106, doi:10.1029/2006JD007111.

Bougeault, P., P. Binder, A. Buzzi, R. Dirks, J. Kuettner, R. Houze, R.B. Smith, R. Steinacker, H. Volkert, 2001: The MAP Special Observing Period. Bull. Am. Meteor. Soc., 82, 433-462.

Rotunno, R., and R. Ferretti, 2001: Mechanisms of Intense Alpine Rainfall. J. Atmos. Sci., 58, 1732-1749.

---

## Referee Comment (RC2) · Anonymous Referee #2 · 22 Jun 2018

Review of paper:

Stucki et al.: Reconstruction and simulation of an extreme flood event in the Lago Maggiore catchment in 1868

General Comments:

The paper aims at the reconstruction of an extreme hydro-meteorological event causing great damages in the Lago Maggiore region and parts of the Swiss Alps, which took place in 1868. The authors combine historical information with a meteorological and hydrological modelling. Their modelling approach makes use of the 20th Century Reanalysis (20CR).

The paper is well written and covers an interesting approach. The authors manage to paint a plausible picture regarding the reconstruction of an at least centennial scale extreme event more than 100 years ago. In principle, this paper fits well in the scope of this journal.

My major concern is that it is not fully clear how specifically the 20CR data are used in their analysis or to derive their high-resolution downscaling simulations. The expected uncertainty of the 20CR data to reproduce a series of mesoscale events is not well addressed. I would guess that the 20CR ensemble is poorly constraint by the few available surface pressure information available at that period. Therefore, I would expect a large ensemble spread and a too smooth ensemble mean over 56 members. The figure they present (section 3.3) show sharp mesoscale feature with respect to streamers, wave breaking or hydrological variables.

In addition, it is not clear to me, how an appropriate forcing for the downscaling is derived from an ensemble mean, since the averaging could induce imbalances in the dynamical fields or at least cause too smooth fields to allow the development of very extreme events. What are the consequences for the high-resolution quantitative results presented? Similar topics are discussed for the representation of windstorms in Welker et al. (2015) paper, which also hints at the smoothing effect of using the ensemble mean.

Both aspects should be more extensively addressed in the paper.

Specific points:

Section 1

I agree with the other reviewer that the introduction is too long and includes aspects which would better fit in the results sections.

Section 2.2 lines 9ff:

How is the initial state of the soil moisture derived? The downscaling period might be too short to derive a reasonably balanced soil.

Section 3.3:

Please state also here which data are used within the chapter – presumably the 20CR ensemble mean.

Figure 6: Not all the variables shown are explained in the caption but discussed later.

Figure 9: There seems to be a typo in the figure caption

Figure 12 right: There is a white curve, which is not explained in the caption

---

## Author Response (AR1)

**Author's Response**

- List of relevant changes
- Replies to Reviewer 1
- Replies to Reviewer 2
- Revised manuscript with tracked changes

**List of relevant changes made in the manuscript**

Added a quantitative assessment of 20CR ensemble mean and members regarding uncertainty: spread, background departure, reduction of ensemble spread by observations, smoothing effect (minimum pressure values).

Included an additional Figure A2 in the Appendix to illustrate the above assessment

Justification of the WRF model setup, description of limitations (e.g. regarding soil moisture equilibrium vs. initialization time, nudging)

Description of limitations regarding the hydrological model (e.g. forecast errors from model parameters, input precipitation, flood peak vs. volume, rain-snow separation)

Description of limitations regarding the E-OBS dataset (e.g. correlation to observations, bias wrt RHiresD dataset)

Moved some specific content from the introduction to the results section

All requested additional references added

Inserted information about the influence of debris flows to flooding processes

Re-introduced observations from Simplon pass in Fig. 4

Added missing information about geopotential height in Fig. 6

Used an interpolation package for visualization of Figs. 6 and 7

Revised graphical elements in Fig. 12

**Reply to Reviewer 1**

**Review of the paper "Reconstruction and simulation of an extreme flood event in the Lago Maggiore catchment in 1868", by P. Stucki et al.**

**General comments**

This paper describes a disastrous hydro-meteorological event that affected Switzerland 150 years ago. Thanks to the availability of the "Twentieth Century Reanalysis" dataset, meteorological and hydrological state-of-the-art models have been applied, to try to obtain quantitative results that complement the documentary/historical data available.

The approach is interesting, and this is probably the earliest event that has been reconstructed/simulated with models, comparing the model results with the earlier observations. The paper is well written and, in my opinion, basically it is worth of publication in NESS. However, I am a bit concerned with the fact that the reanalysis at that time is based on very scarce data (essentially sporadic surface observations) if compared to nowadays situation, so that it is affected by large uncertainties (particularly at small scales and in atmospheric moisture content) that may adversely reflect on the simulations, especially those at high resolution. The latter can be useful since they provide more accurate description of the orography and land properties and of the atmospheric dynamics, but one cannot rely too much upon detailed results of the model simulations: this limited predictability problem at the convective scale severely affects the quantitative precipitation forecasts even in our days. I can only imagine how severe it is at the time of the event! Therefore, I think that the authors should devote more space to try to quantify or at least describe qualitatively if not quantitatively the uncertainties of their results.

Thank you for the appreciation of language, style and content. We are pleased to read that you judge the manuscript worth publishing after revision.

We understand the concerns of both reviewers about inherent uncertainty in 20CR, which might be passed on to the regional model, and the limited predictability of parameters like precipitation. Similarly, we understand the concern about a large spread and smoothing effects. In the following, we respond to both reviewers regarding these concerns.

In an ideal world, we would assess all uncertainties and include an 'uncertainty bar' for each step, as it would be desirable for a risk assessment study. Unfortunately, this is far beyond the possibilities of this study. Yet, we clearly see the need for describing the limitations and uncertainties of the used tools and the obtained results. In the revised version of the manuscript, we do this for each step of the modeling chain in the respective results sections.

Most prominently, we have added an extended quantitative assessment regarding the 20CR ensemble mean and members in Sect. 3.3. First, we address the spread of the 20CR SLP fields over Central Europe, which is mostly below 1 hPa. We show that the isobars remain within a narrow band, and that the ensemble mean represents a middle scenario; it runs always within the bulk of the members. To further assess the uncertainty of 20CR in the region and time of interest, we analyze specific diagnostics for each observation that went into the assimilation. Specifically, we analyze the ensemble spread of the first guess and of the analysis, and we analyze the background departure (obs-bck). Our analysis shows that the observations reduce the ensemble spread by a factor of 0.5 to 0.75. Observation departures are relatively small and no clear pattern appears. In addition, we analyze the smoothing effect by comparing minimum pressure in the ensemble mean to the minimum pressure values from the full ensemble. The minimum pressure values in the ensemble mean lie between the 54th and 57th percentile of the ensemble members. This illustrates that there is hardly any smoothing effect in the pressure fields of the ensemble mean for our region of interest.

The corresponding analyses are shown in Fig. A2 in the Appendix. The analyses support the choice of the ensemble mean as a plausible and realistic depiction of the meteorological situation in 1868.

We also include descriptions of limitations in the subsequent steps of the model chain. Please refer to the specific comments for more detailed information. In summary, the limitations of the weather and hydrological model setups are described in a semi-quantitative or qualitative way. The choice of the WRF setup is now justified at the top of Sect. 3.4. Uncertainties regarding the hydrological modeling are addressed in a separate paragraph in Sect. 3.5. Specifically, the description of uncertainties regarding E-OBS are included in the data section.

Your general comments had us also think about wha t the choices of tools and model setups mean and what the simulation outcomes represent under these circumstances. Following Compo et al., (2011), we regard the ensemble mean as a 'minimum-error estimate', and the results from the hydro-meteorological simulations can be regarded as a well-reasoned proposal of how the weather evolved during the 1868 event. These notions are elaborated in the article, and also appear in the conclusions.

We hope that we meet the expectations of the reviewers with these revisions and adaptations.

.

**Punctual comments**

The Introduction appears a bit too long and detailed – it contains some treatment that should be postponed to the specific chapters.

According to your suggestion (and the same by reviewer 2), we have moved some specific content to the results sections, i.e. concerning the Swiss legislation after the 1868 event, the mesoscale flow modulation on the south side of the Central Alps, and the hydrological response. Because this is an interdisciplinary study, we think we should shortly introduce some important concepts, e.g., political responses, or the role of IVT, PV-streamers, or the zero-degree line.

Page 1, line 36: if AS stands for Central Alps, it should be CA. But in the context of the paper I had the suspect that it means "Southern Alps" (or Alps South?). I suggest, in any case, to use initials that correspond to the English terms.

We have changed it to "SA", which now stands for "south side of the (Central) Alps"

Page 2, lines 1-7: the paper Malguzzi et al (2006) (see also below "References") should be referenced here, with some short description, because it is very similar to the present paper in various aspects: it describes a major historical (1966) flood event that affected the (Eastern) Alps (and not only – it was considered the century flood in Italy). It applies a very similar approach (meteorological and hydrological model chain), although the used reanalysis (and also the verification) takes advantage from a much better data coverage than available one century before!

This is indeed a very useful article with a number of similarities to our study. We use it as a reference in the introduction and at several places in the article. Here, we speak about the particular floods in 1993 and 2000, this is why it is not included at this stage.

Page 2, line 9: LMR: Lago Maggiore Region?

Yes; the information is added for clarification.

Pages 2-3: somewhere in the Introduction (and not only at page 12), the MAP international project must be mentioned (for example quoting Bougeault et al, 2001): its major objective was to study (from the observational and modelling point of view) the atmospheric processes related

to heavy precipitation and flood the Alps, with its largest observational effort concentrated just in the Lago Maggiore area.

Thank you for the reference. We added it in the Introduction in the context of heavy precipitation on the south side of the Alps.

Page 3: lines 5-18: regarding the PV streamers west of the Alps, I think that the main interesting aspect (related to the orographic forcing) is that the (orographic) precipitation occurs more to the east, with respect to the position of the PV anomaly, than expected in the case of flat terrain. In other terms and using more traditional synoptic concepts, while in the flat case precipitation has to be expected ahead of the cold front, more or less in the area of the warm conveyor belt, in the case of the Alps (or similar orography perpendicular to the more-or-less southerly flow) precipitation may be heavy also in the warm sector well in advance of the cold front. It must be remembered, in any case, that the orography can change the synoptic scale flow at scales larger than those of the orography itself.

Thank you for the comment, this is a very interesting hypothesis. We decided not to change the text for the following reason: The only paper to our knowledge that systematically looked at the link between WCBs and heavy precipitation on the south side of the Alps finds that a substantial fraction of the heavy precipitation events are co-located with a WCB:

Pfahl, S., E. Madonna, M. Boettcher, H. Joos, and H. Wernli, 2014: Warm Conveyor Belts in the ERA-Interim Dataset (1979-2010). Part II: Moisture Origin and Relevance for Precipitation. Journal of Climate, 27, 27-40.

And we therefore think that we would need to investigate first in more detail if these WCBs are at an unusual location, i.e. further east in the warm sector, before writing such a statement with confidence. We think that this would be a very interesting research question to address although beyond the scope of this paper.

Page 4, lines 19-25: please give some more info here about the 20CR for those not familiar, in particular for what concerns the input observations (other variables besides surface pressure? T and humidity?), the available variables on the pressure surfaces and at the ground) and the degree of uncertainty as estimated in the literature.

We added some information about 20CR here. As mentioned above, we treat the uncertainty separately in the results section.

Page 5, lines 5-8: the application of the nudging, although justified by the need of constraining the forward meteorological model to run close to the reanalysis "trajectory", has drawbacks that should be mentioned: for example, this is not a real "hindcast" experiment – how the precipitation forecast differs from a pure forward integration, i.e. without nudging? (one test should be made at least for one case).

Unfortunately, running a new simulation without nudging is beyond our capacities at this stage. We are currently running such experiments for a potential future study on a more recent event with complicated flow. Nevertheless, we think we can justify the nudging in the specific setup for the 1868 case with the necessity not to let the simulation run too freely. This is now mentioned, and we also mention that the results cannot be regarded as stemming from a pure hindcast experiment.

Page 5, lines 21-25: please explain better the procedure – include some words about the quality/limitations/uncertainties of the E-OBS dataset.

The according paragraph has been extended and now includes some information about the quality

/ uncertainty.

Page 6, lines 15-17: there is similarity with the 1966 event described in Malguzzi et al: in both cases there is strong enough precipitation at the divide and on the downstream side of the Alps to cause flooding also of the rivers flowing on the north side. It is not clear, however, to what extent such precipitation "originates" on the upstream side (due to transport of cloud condensates and hydrometeors across the Alpine crest) or on the downstream side (including possible thunderstorms). Model results (see Fig. 8) do not seem to represent this aspect.

We agree on this comment. In the article, we mention the transport of condensates with stronger winds (e.g. P19, L7ff) and that the model does not fully capture the precipitation patterns and intensites on the north side of the Alps.

Page 6, line 24: it is not clear if only the "inflation" of the Swiss Franc is considered here or also the fact that the nowadays (economic) damages would be much larger due to the real value and vulnerability of today's infrastructures, resources etc. It seems to me the first is the case here, but the second would be more interesting...

It is inflation only; we added an according clause for clarification, thank you.

Page 8, lines 11-14: I think that the most dramatic aspect, which makes this event so exceptional, is the occurrence of an entire sequence of heavy precipitation cases. Of course, this aspect cannot be "explained" in terms of atmospheric dynamics, even with the help of model simulations.

We agree that the repeated occurrence of similar patterns was decisive for the extreme magnitude of the event. We emphasize this finding in Sect. 3.3 and 4.

Page 8, line 37 (and elsewhere, for ex. page 9, lines 25-26): the expression "Rossby wave breaking" is used several times, but I think that not only it is too technical, it is also ambiguous and should deserve an explanation in more traditional terms.

We agree that for an interdisciplinary audience, this might be very technical. We omit the term in the first occurrence and replace it with a less technical explanation, because we do not actually show the according panel:

[Figure]

In the second occurrence, we leave the term because it is a characteristic feature for such events on the south side of the Alps and shows that even with the limitations regarding resolution and assimilated data, 20CR is capable of detecting such complicated weather variables in a physically consistent way. For the non-specialist reader, we add an explanation of how the breaking manifests in the weather chart (Fig. 6b.)

Page 9, lines 2 and 15; page 10 line 1: a blocking anticyclone (in the traditional meteorological literature) is something different (in brief, an anticyclone located at lat. 50-70 over the Atlantic, Pacific or northern Europe, deviating the westerlies for several days). A ridge downstream of a trough is (I believe) a component of the same Rossby wave, that can become almost stationary. So I think that it not very correct to say that the ridge blocks the easterly propagation of the system.

We deleted this sentence: Until 17 September 1868, a downstream stationary ridge prevented eastward propagation of the system (Figure 6a).

Page 9, line 14: I guess "high-PV", not "low-PV".

high PV trough - thank you for pointing this out

Page 9 line 17: perhaps an "atmospheric river"?

Indeed, this feature resembles an atmospheric river. However, the term is mostly used for structures that are >1500 km long and have constant IVT values of >350 kg m$^{-1}$ s$^{-1}$. Hence, we prefer not to use the term in our specific context.

Page 9, lines 27-33: qualitatively similar results have been obtained from trajectory computations by Bertò et al, 2004 (among others), for similar events of heavy precipitation on the southern side of the Alps. I suggest to quote this paper (see below the full reference).

Thank you for this reference; their analysis greatly supports our results and we are happy to include it.

*Page 9, lines 36-37: I do not think there is a "plausible explanation" – after all, it is not given here.*

We can see that this term might be misleading here. The sentence is therefore rephrased.

*Page 10, lines 5-8: some more clear should be provided here (or above) about the uncertainties of 20CR.*

Two paragraphs and a figure (in the Appendix) are added at the top of Sect. 3.3. See also our reply to your general comments.

*Section 3.4 is a bit too speculative, given the uncertainties as mentioned above (see the major comments). If uncertainties affect the low resolution, they can only be larger at higher resolution.*

Our line of thought is a little different here. In Sect. 3.3 we come to the conclusion that the ensemble mean is a valid, well-reasoned choice for further analyses (see our reply to your general comment about uncertainty and the reply to the comment on uncertainties in 20CR above). The same, we argue that the chosen model setup offers a valid, well-reasoned proposal of how the weather evolved on a local scale during the 1868 event. The quality of this proposal can then be assessed by comparisons to observations and to analyses of modern analog cases.

*Page 10, lines 31 and 33: "well2 and "very good" seem too optimistic, at least in absolute terms (for both amount and spatial distribution of precipitation). I agree that the simulation provide useful meteorological information, perhaps better than expected, but I would use the word "satisfactory" rather than "very good".*
We can agree on changing to 'satisfactory' here.

*Page 11, lines 7-9: these aspects were investigated during the MAP field experiment (as mentioned below). Here I would quote the paper by Rotunno and Ferretti (2001), providing an interesting explanation of the enhanced convergence due to horizontal inhomogeneities of moisture content at low levels in the incident flow.*

This is indeed an interesting investigation, which nicely supports our study, although we do not explicitly look at horizontal differences of moisture content in the low-level flow. We added the reference in the context of the MAP field experiment in the discussion part of Sect. 3.4.

*Page 11, line 14: quite typical when convective scale are treated explicitly!*

Maybe there is a misunderstanding here. We revised the sentence to:
All these dynamics are consistent with ...

*Page 11, line 35: wouldn't it be simpler saying "relatively low relative humidity".*
We can see your point, and had discussed it internally before submitting. In conclusion, we would like to keep the current phrasing because the spread is what we actually see in the skew-T diagram.

*Page 12, line 32: drop "were".*

This was dropped in the final submission.

*Page 13, regarding the effects of forests: I agree that forests may not mitigate flooding, however I wonder (but I am not an expert!) if they may act to reduce landslides (a very important cause of casualties and damages - see page 6, line 25) and reduce the amount of mud in flooded water that may worsen the effects.*
Thank you for pointing out this aspect. We have added some information about the influence of debris flows.

 I think that the moisture flux is not a "precursor" of precipitation like the other factors, since it is almost contemporary to precipitation.

This is correct, thank you for the hint. To keep the metaphor, we added 'and companions'.

Page 14, line 12: "very well": again too optimistic and not consistent with the results presented here.

We changed the phrasing to 'are compatible with'.

**Figures**

Fig. 4: the simulated precipitation seems rather low compared to the observed one. Note also that the Simplon station used to sample the simulation is not included in the station list for the observations – why?

Yes, this underestimation of precipitation is mentioned in Sect. 3.4.
The Simplon observations had been lost somewhere in the process because they cointain NA's. This is corrected now, thank you for the closer look.

Fig. 6 and 7 are not very clear. This is due mainly to the discontinuous (boxes) coloured fields (shade). This should be avoided using an interpolating graphical package.

The original shade was used to show the grids of 20CR. However, we can see that this would be at the expense of clarity and readability of the figure. Therefore, we adapted the figures according to the request.

Fig. 7: panel d) suggests the presence of an "atmospheric river".

See the comment above: "Indeed, this feature resembles an atmospheric river. However, the term is mostly used for structures that are >1500 km long and have constant IVT values of >350 kg m$^{-1}$ s$^{-1}$. We prefer not to use this term."

Fig. 9a and 9b: it is very difficult to appreciate the differences/similarities with the observations of fig. 5. Perhaps the above figures should be enlarged, plotting on them the observations of fig. 5 with circles.

Thank you for your suggestion. We have indeed tested a number of options for mapping the precipitation obtained from observations and the WRF simulation. One of the options was an overlay map for Fig. 5 (see below). In the end, we decided to refrain from it, for several reasons. We would like to emphasize the aspect of reconstruction and give appreciation to the labor-intense research and digitizing with a stand-alone figure. We would break (and spoil) the narrative of the study, which goes from traditional reconstructions to numerical simulations. An overlay would hide important detail, even with transparent colors. As a workaround, the interested reader may zoom into the two figures side by side on the computer screen.

[Figure]

General Comments:

The paper aims at the reconstruction of an extreme hydro-meteorological event causing great damages in the Lago Maggiore region and parts of the Swiss Alps, which took place in 1868. The authors combine historical information with a meteorological and hydrological modelling. Their modelling approach makes use of the 20th Century Reanalysis (20CR).

The paper is well written and covers an interesting approach. The authors manage to paint a plausible picture regarding the reconstruction of an at least centennial scale extreme event more than 100 years ago. In principle, this paper fits well in the scope of this journal.

My major concern is that it is not fully clear how specifically the 20CR data are used in their analysis or to derive their high-resolution downscaling simulations. The expected uncertainty of the 20CR data to reproduce a series of mesoscale events is not well addressed. I would guess that the 20CR ensemble is poorly constraint by the few available surface pressure information available at that period. Therefore, I would expect a large ensemble spread and a too smooth ensemble mean over 56 members. The figure they present (section 3.3) show sharp mesoscale feature with respect to streamers, wave breaking or hydrological variables.

In addition, it is not clear to me, how an appropriate forcing for the downscaling is derived from an ensemble mean, since the averaging could induce imbalances in the dynamical fields or at least cause too smooth fields to allow the development of very extreme events. What are the consequences for the high-resolution quantitative results presented? Similar topics are discussed for the representation of windstorms in Welker et al. (2015) paper, which also hints at the smoothing effect of using the ensemble mean.

Both aspects should be more extensively addressed in the paper.

Thank you for the appreciation of language, style and content, and are pleased to read that you judge the manuscript worth publishing after revision.

We understand the concerns of both reviewers about inherent uncertainty in 20CR, which might be passed on to the regional model, and the limited predictability of parameters like precipitation. Similarly, we understand the concern about a large spread and smoothing effects. In the following, we respond to both reviewers regarding these concerns.

In an ideal world, we would assess all uncertainties and include an 'uncertainty bar' for each step, as it would be desirable for a risk assessment study. Unfortunately, this is far beyond the possibilities of this study. Yet, we clearly see the need for describing the limitations and uncertainties of the used tools and the obtained results. In the revised version of the manuscript, we do this for each step of the modeling chain in the respective results sections.

Most prominently, we have added an extended quantitative assessment regarding the 20CR ensemble mean and members in Sect. 3.3. First, we address the spread of the 20CR SLP fields over Central Europe, which is mostly below 1 hPa. We show that the isobars remain within a narrow band, and that the ensemble mean represents a middle scenario; it runs always within the bulk of the members. To further assess the uncertainty of 20CR in the region and time of interest, we analyze specific diagnostics for each observation that went into the assimilation. Specifically, we analyze the ensemble spread of the first guess and of the analysis, and we analyze the background departure (obs-bck). Our analysis shows that the observations reduce the ensemble spread by a factor of 0.5 to 0.75. Observation departures are relatively small and no clear pattern appears. In addition, we analyze the smoothing effect by comparing minimum pressure in the ensemble mean to the minimum pressure values from the full ensemble. The minimum pressure values in the ensemble mean lie between the 54th and 57th percentile of the ensemble members. This illustrates that there is hardly any smoothing effect in the pressure fields of the ensemble mean for our region of interest. The corresponding analyses are shown in Fig. A2 in the Appendix. The analyses support the choice of the ensemble mean as a plausible and realistic depiction of the meteorological situation in 1868.
We also include descriptions of limitations in the subsequent steps of the model chain. Please refer to the specific comments for more detailed information. In summary, the limitations of the weather and hydrological model setups are described in a semi-quantitative or qualitative way. The choice of the WRF setup is now justified at the top of Sect. 3.4. Uncertainties regarding the hydrological modeling are addressed in a separate paragraph in Sect. 3.5. Specifically, the description of uncertainties regarding E-OBS are included in the data section.

Your general comments had us also think about what the choices of tools and model setups mean and what the simulation outcomes represent under these circumstances. Following Compo et al., (2011), we regard the ensemble mean as a 'minimum-error estimate', and the results from the hydro-meteorological simulations can be regarded as a well-reasoned proposal of how the weather evolved during the 1868 event. These notions are elaborated in the article, and also appear in the conclusions.

We hope that we meet the expectations of the reviewers with these revisions and adaptations.

**Section 1**

I agree with the other reviewer that the introduction is too long and includes aspects which would better fit in the results sections.

According to your suggestion (and the same by reviewer 1), we have moved some specific content to the results sections, i.e. concerning the Swiss legislation after the 1868 event, the mesoscale flow modulation on the south side of the Central Alps, and the hydrological response. Because this is an interdisciplinary study, we think we should shortly introduce some important concepts, e.g., political responses, or the role of IVT, PV-streamers, or the zero-degree line.

**Section 2.2 lines 9ff:**

How is the initial state of the soil moisture derived? The downscaling period might be too short to derive a reasonably balanced soil.

The soil moisture in the WRF simulations is initialized from the reanalysis soil moisture data, and indeed, the soil moisture is a long-memory variable that reaches its equilibrium later than the atmospheric variables. However, the relatively short spin-up time used here (6-12 hours) allows for a good reproduction of the event (see Messmer et al., 2017, using a spin-up of 6 hours for a comparable experiment). Overall, the model spin-up time has been determined in a trade-off between avoiding model adjustment and remaining faithful to the reanalysis data. We have inserted an according sentence in the article.

**Section 3.3:**

Please state also here which data are used within the chapter – presumably the 20CR ensemble mean.

Yes, this is an essential information. We have also added an extended explanation of why we chose the 20CR ensemble mean; see our reply to the general comments.

Figure 6: Not all the variables shown are explained in the caption but discussed later.

Thank you for the hint. We have inserted the missing information about geopotential height.

Figure 9: There seems to be a typo in the figure caption

Yes, the caption is adapted. Thank you for pointing this out.

Figure 12 right: There is a white curve, which is not explained in the caption

Thank you for the hint; the figure has been revised and replaced.

[revised manuscript text omitted]

**3.3    Synoptic-scale atmospheric conditions**

The synoptic-scale atmospheric conditions are analyzed using the 20CR dataset. A particular feature of 20CR is that it presents a range of potential solutions resulting from the uncertainty of the measurements and the distribution of the assimilated data. This range is represented by an ensemble of 56 members, all of which are equally likely. The range of outcomes from analyzing the ensemble members can be exploited for risk assessment studies (e.g. Welker et al., 2015). For studies that do not have the objective or the means for such a procedure, like the present one, the ensemble mean is a natural and preferred choice (e.g. Caillouet et al., 2016; Carillo et al., 2017; Michaelis and Lackmann, 2013; Parodi et al., 2017).

In our case, the selection is based on comparisons of pressure fields in the 20CR ensemble mean and members. Figure A2 depicts all 56 solutions for the 1008-hPa isobars and for four time steps during the 1868 event. Over Central and South-Western Europe, the isobars remain within a narrow band. The ensemble mean represents a middle scenario; it runs always within the bulk of the members. This small bandwidth is also manifest in the standard deviation (the "spread" in terms of 20CR; see Compo et al., 2011) of the pressure fields, which is mostly below 1 hPa in the region of interest for all time steps. At the three points in time where the low-pressure system is located over Central Europe, the pressure minima of the 56 ensemble members are mostly co-located within a small number of grid points, and the interquartile range spans 5 hPa or less. The minimum pressure values in the ensemble mean lie between the 54[th] and 57[th] percentile of the ensemble members. This illustrates that there is hardly any smoothing effect in the pressure fields of the ensemble mean, that is, over the region of interest. In contrast, the uncertainties become much larger over Africa or the North Atlantic. On 28 September 1868, for instance, the low-pressure system is only well-defined along the European continent.

To further assess the uncertainty of 20CR in the region and time of interest, we analyzed specific diagnostics for each observation that went into the assimilation (see the ISPD; Cram et al., 2015). Specifically, we analyzed the background departure (observation minus first guess). Figure A2 shows that the observations reduce the ensemble spread by a factor of 0.5 to 0.75 at almost all locations. Observation departures are relatively small and no clear pattern of deepening or weakening the pressure fields appears. This means that there are no outlier weather stations that dominate the field.

Overall, the relatively small uncertainty ranges in our analyses show that the 20CR ensemble mean is able to deduce the large-scale physical parameters for this area of the world and for our case in a physically consistent way. As such, the 20CR ensemble mean represents a "minimum-error estimate" of the true state (following Compo et al., 2011). Of course, with so little available station data for 1868, this "minimum error" may be large in comparison to more recent cases. Still, it is the best we have to date while improvements of the 20CR are on the way. In conclusion, we regard the 20CR ensemble mean as a well-reasoned and valid choice for further analyses. In the following, only the ensemble mean is used for synoptic-scale analyses and dynamical downscaling.

HereTo start with, we employ a PV- and IVT-perspective to look at the upper-level dynamics during all four episodes of heavy precipitation. The first episode (episode 09-17; see Sect. 3.2) was characterized by the amplification of an upper-tropospheric wave (i.e. a meridionally undulating upper-level wind field) and

the development into a PV-streamer that reached northern Africa (on the 330-K isentrope) on 16 September 1868 (not shown).  During this time, this trough and a subsequent trough moving in from upstream merged into one broad trough associated with a very strong moisture flux reaching the SSA from the south-west (Figure 6a, c and e).

At the beginning of the second episode (09-22), another trough formed over western Europe on 21 September 1868 and cyclonical Rossby wave breaking occurred on 22 September 1868 (detectable as an S-form in the PVU contours over the western Alps in Fig. 6b). The associated, very strong south-westerly flow brought moist air towards the SSA (Figure 6d and f). A strong ridge, located downstream of the trough, persisted through 23 September 1868.

[revised manuscript text omitted]

**3.4   Meso-scale atmospheric conditions**

Given the coarsely resolved 20CR dataset, the analyses above also cannot explain local phenomena like the observed patches of precipitation in the LMR and across the Alpine divide. For this, we use the regional weather model WRF. A large number of setups are available. For instance, simulated precipitation patterns may differ according to the choice of microphysics schemes (e.g. Pieri et al., 2015, and references therein). For our study with restricted resources, most parameters are as in an operational setup for weather forecasting, with the exception of large-scale nudging to keep the model in basic agreement with the reanalysis (see Sect. 2.2). Hence, the simulation output must be regarded as a well-reasoned proposal of how the weather evolved on a local scale during the 1868 event. Verification and validation of the simulation results is made in two ways: by comparisons of the simulation output to observations and eye-witness reports where available, and by comparisons of the simulated weather patterns and dynamics to analyses of comparable modern events.

[revised manuscript text omitted]

Regarding our region of interest, three sources of uncertainty in hydrological modeling were quantified by Zappa
32   et al. (2011) for flood forecasting in the Verzasca basin, a contributory river to Lago Maggiore. They found that the spread of the forecast attributed to the hydrological model parameters is four times smaller than the contribution of uncertain rainfall fields by the weather radar; numerical weather forecast has by far the largest contribution to the forecast spread. In addition, they found a very small contribution of flood-peak simulations to the overall
36   uncertainty. This low sensitivity to initial conditions is owed to the fact that soils are rather shallow in this area, thus rapidly saturated and prone to runoff generation. We infer that for the 1868 extreme event with intense runoff generation, the flood peak is not very sensitive to uncertainties of the model parameters. Furthermore, one should consider that the key factor for the Lago Maggiore flood was rather the flood volume than the peak – the volume

is far less parameter-dependent. One source of uncertainty to affect the flood volume might be the separation between snow and rain. This separation was not relevant in 1868 as the snow line was located well above 3000 m a.s.l (Fig. 11). We conclude that for this Lago Maggiore flood, the uncertainty from the rainfall input is by far larger (orders of magnitude) than from model parameters.

-The simulations show that the flood event followed a rather dry to normal late-summer period, and a first very wet spell during September. During this wet period, soil storage capacity was reached and the Lago Maggiore lake level rose significantly. For the second wet period (comprising the episodes 09-27 and 10-03), WRF simulations are added to the analog-driven simulations. Differences appear between the analog- and the WRF-derived precipitation: WRF shows a decreasing intensity for the four major rainfall events, while the analog method shows increasing intensities (Figure 12, left panel). Total precipitation sums are approximately equal for both approaches. The simulated lake level responds accordingly (Figure 12, right panel), i.e. with a significant increase in the lake level during the early treatment period in the WRF simulation, and a smaller increase later (and vice versa for the analog method).

Although our hydro-meteorological model chain tends to overestimate the lake levels (see Sect. 2), we are not able to reproduce the observed lake level peak. Possible reasons are manifold. Firstly, the reconstructed precipitation input could have been strongly underestimated. However, given the very steep rise in the observed lake level, this precipitation amount would have been enormous. Secondly, a damming of the Lake due to a log jam could also have occurred. We evaluated an idealized model experiment of such a log jam by reducing the outflow amount to one third of the theoretical relationship for the time of the steepest increase. At the time of the highest peak, such a log jam at the Sesto Calende bridge would have needed to be resolved in one day to match the steep decline observed in the lake level records. However, to our knowledge, such a log jam is unreported in the historical documents. Thirdly and most likely, the reason for the strong underestimation is due to changes in the lake level–outflow relationship. Ambrosetti (1994) reviewed long-term lake level changes in the 19[th] and 20[th] centuries and stated that the high lake levels recorded before 1868 are unlikely to be achieved again, as a ridge on the lake outlet was heavily eroded during the 1868 flood event (minus 30 cm). As we do not have any hint of the shape of this ridge and the cross-section at the lake outflow, we cannot prove this theory. However, we assume that changes in the lake level–outflow relationship is the most likely reason, among the ones discussed.

[Figure]

**Figure 12. Hydrological simulation of the 1868 flood event in the LMR. The left panel displays the precipitation input to the system (area averages in mm) for the WRF downscaling simulation (positive values) and the analog resampling method (negative values), as well as mean areal soil storage. The right panel shows the observed (blue lines) and simulated (colored lines) lake levels from August to October 1868. WRF scenarios are only simulated for the treatment period shaded in grey. The deforestation scenario was computed for a full hydrological year from 1 October 1867 to 31 October 1868.**

Furthermore, we estimate the effect of deforestation on the flood event in 1868. Our simulations show only very small differences, indicating that even a higher percentage of forest cover (conditions similar to today) would not have dampened the impacts of such an extreme event. This is mainly due to the limited and rapidly filled storage capacity of the vegetation, and especially of the mostly very shallow soils (Figure 12, left panel). This is confirmed by literature values regarding forest canopy and forest floor interception capacity, which are about 3 mm at maximum for beech (*fagus sylvatica*; Gerrits et al., 2010) and 4.7 mm at maximum for spruce (*picea abies*; Breuer et al., 2003). In relation to a total precipitation input of 280 mm, the limited significance of forest cover during such extreme events is clear. In addition, we assume the effect of debris flows to be negligible in our case. Clearly, deforestation increases the disposition for debris flows in steep terrain. They could block river discharge, with severe ramifications due to the impounding of water and an often sudden release of blocked water (Badoux et al. 2014). However, these processes happen on an hourly time scale (Borga et al. 2014), while the rise of the Lago Maggiore lake level took several days.

**4      Summary and conclusions**

In this study, we have combined traditional reconstructions and numerical simulations to assess an extreme flood that occurred in the Lago Maggiore catchment and surrounding areas on the south side of the central Alps during the end of September and early October 1868. We have taken an interdisciplinary approach to describe damage and impacts, the flood-inducing precipitation, the driving atmospheric conditions, and potential reasons for the extreme flood levels.

The analyses show that the systematic use of contemporary sources, such as documentaries, weather annals, and eye-witness and administrative reports, results in a very detailed picture of the impacts and damages on regional and even municipality levels. Damage was concentrated around Lago Maggiore and south of the Alpine divide, but also reached across the Alpine divide into Graubünden, Valais, and Central Switzerland.

The synoptic (large-scale) atmospheric conditions have been analyzed using 20CR, a global reanalysis dataset. The ensemble mean is chosen as a minimum-error estimate for further analyses based on low spread of the ensemble over the region of interest and a small smoothing effect in the pressure fields. The synoptic situation is comparable to modern cases. However, we additionally find a repeated occurrence of similar patterns over four distinct episodes from 17 September to 4 October 1868. Typical precursors and companions of heavy precipitation are found with PV-streamers, atmospheric wave breaking, and persistent, very high (IVT >300 kg m$^{-1}$ s$^{-1}$) southerly moisture fluxes towards the Alps.

Simulated weather variables at meso- (local) scales are obtained from dynamical downscaling 20CR input to a 2-km grid using the limited-area weather model WRF. Given the uncertainties stemming from the model setup and the coarse initial and boundary conditions, we regard the simulation output as a well-reasoned proposal of how the weather evolved during the 1868 event. However, we conclude from a range of comparisons to observations and modern analog cases that the downscaling experiment delivers very plausible results. Concretely, the simulated precipitation rates, patterns, and atmospheric dynamics are compatible with the observed precipitation and damages. They also agree well with existing concepts of forced moisture convergence in the concave orography around Lago Maggiore, and subsequent uplift, convection and repeated north-eastward propagation of convective cells. Hence, the WRF model is able to reproduce small-scale phenomena of an event that occurred in 1868, and to achieve a realistic cloud-resolving simulation with downscaling from 20CR.

In addition, we simulate freezing levels that are mostly close to 4000 m a.s.l., meaning that virtually none of the heavy precipitation was stored as snow and delayed runoff. The subsequent hydrological modelling captures the gradual increase in the Lago Maggiore water level. The simulated and observed peak levels differ by approximately 2 m. Reasons for this gap may be found in unknown riverbed profiles at the lake outflow. Reduced forest cover in the Lago Maggiore catchment at that time did not have an influence on the extreme response of Lago Maggiore; experiments with different percentages of forest cover have not resulted in different responses in the lake level.

This is in contrast to leading forestry policies at that time, which presumed a forest storage effect. Based on studies of French engineers from the late 18$^{th}$ century, the idea of a general correlation between forest cover and floods was established in Switzerland in the mid-19$^{th}$ century and strongly supported by the Swiss forestry association. The catastrophic flood in 1868 triggered the implementation of afforestation programs and the construction of hydraulic structures. Following the event, the federal government enacted forest and hydraulic engineering laws, and committed to financially supporting protection measures. As a result, many rivers were channeled, torrent controls established, and alpine meadows afforested in the following decades. A differentiated consideration of the role of forest cover as a flood protection measure has only taken place since the late 20$^{th}$ century, and the paradigm has only recently shifted towards re-naturalization of waterways.

In conclusion, we find – for this specific case – very good agreement between traditional reconstructions and numerical simulations on almost all levels, mind you for an event in the mid-19$^{th}$ century. To our knowledge, the provided local-scale weather maps are the earliest to date, but we hope that this is not the case for long. Indeed, these findings highlight the broad prospects for these combined analyses with respect to studies of weather events

and their impacts back to the early 19th century, and maybe beyond. They also highlight the opportunities for cross-validations between scientific disciplines, be it meticulously examined documents for validating weather or runoff models, or numerical support for historical climatology studies.

**Acknowledgements**

The work was supported by the Oeschger Centre for Climate Change Research and by the Swiss National Science Foundation (project CHIMES). We would like to thank Ralph Rickli, Christian Rohr and Martin Stuber for fruitful discussions about weather dynamics and historical aspects. We also thank Erin Gleeson for the careful language editing. Support for the Twentieth Century Reanalysis Project version 2c dataset is provided by the U.S. Department of Energy, Office of Science Biological and Environmental Research (BER) program, and by the National Oceanic and Atmospheric Administration Climate Program Office.

**Appendix**

[Figure]

**Figure A1: Validation of the hydrological model performance by means of long-term average (1960–2010, left panel) and daily value-based Q-Q representation, indicating a fair model performance with the tendency to overestimate higher lake levels. In the left panel, we show the model representation assuming a simple winter storage regulation (red line).**

[Figure]

**Figure A2: Mean sea level pressure (hPa) calculated from 20CR for (a) 17 September 1868 12 UTC (b) 22 September 1868 12 UTC (c) 27 September 1868 12 UTC and (d) 3 October 1868 12 UTC. The background color shade indicates the standard deviation ("spread") of the ensemble (hPa). Light grey contours indicate the 1008-hPa isobars for all 56 ensemble members. Bold black contours show the isobars in the 20CR ensemble mean. Grey dots mark the locations of pressure minima for all ensemble members within the inset box; the darker the dot, the more members have the minimum pressure at this grid point. The boxplot in the lower left corner of each panel indicates the median, interquartile range and full range of the ensemble pressure minima; the grey diamond indicates the pressure minimum in the ensemble mean. From top to bottom, the values to the left of the boxplot indicate the position of the ensemble mean w.r.t. the according percentile (%) of the ensemble members, the interquartile range and the full range of the ensemble members in hPa. Triangles mark the locations of the pressure observations that are available for assimilation into 20CR from 06 UTC to 12 UTC at the specific date (ISPD; Cram et al., 2015). The adjustment effect (magnitude qualitatively shown by the size of the triangle) by the pressure measurements towards weakening (deepening) of the surrounding pressure fields is shown with up-pointing (down-pointing) triangles; the color shade represents the ratios of the ensemble spread in the analysis w.r.t. the first guess forecast (see also Compo et al., 2011).**

---

## Referee Report (RR1)

**Review of the manuscript nhess-2018-134-version4**

The authors addressed all issues raised by the reviewers. Their argumentation regarding these issues and their choices is satisfactory. They were able to underlay their reasoning with sound arguments and additional analyses. Especially, their new Figure A2 is impressive, as it contains a huge amount of information, with respect to the uncertainty of the 20CR ensemble and is still being well comprehensible (with large magnification on a computer screen). I recommend this manuscript for publication as is.